# ROBUST ACTIVE DISTILLATION

**Cenk Baykal, Khoa Trinh, Fotis Iliopoulos, Gaurav Menghani, Erik Vee**
Google Research
`{baykalc,khoatrinh,fotisi,gmenghani,erikvee}@google.com`

## ABSTRACT

Distilling knowledge from a large teacher model to a lightweight one is a widely successful approach for generating compact, powerful models in the semi-supervised learning setting where a limited amount of labeled data is available. In large-scale applications, however, the teacher tends to provide a large number of incorrect soft-labels that impairs student performance. The sheer size of the teacher additionally constrains the number of soft-labels that can be queried due to prohibitive computational and/or financial costs. The difficulty in achieving simultaneous *efficiency* (i.e., minimizing soft-label queries) and *robustness* (i.e., avoiding student inaccuracies due to incorrect labels) hurts the widespread application of knowledge distillation to many modern tasks. In this paper, we present a parameter-free approach with provable guarantees to query the soft-labels of points that are simultaneously informative and correctly labeled by the teacher. At the core of our work lies a game-theoretic formulation that explicitly considers the inherent trade-off between the informativeness and correctness of input instances. We establish bounds on the expected performance of our approach that hold even in worst-case distillation instances. We present empirical evaluations on popular benchmarks that demonstrate the improved distillation performance enabled by our work relative to that of state-of-the-art active learning and active distillation methods.

## 1 INTRODUCTION

Deep neural network models have been unprecedentedly successful in many high-impact application areas such as Natural Language Processing (Ramesh et al., 2021; Brown et al., 2020) and Computer Vision (Ramesh et al., 2021; Niemeyer & Geiger, 2021). However, this has come at the cost of using increasingly large labeled data sets and high-capacity network models that tend to contain billions of parameters (Devlin et al., 2018). These models are often prohibitively costly to use for inference and require millions of dollars in compute to train (Patterson et al., 2021). Their sheer size also precludes their use in time-critical applications where fast decisions have to be made, e.g., autonomous driving, and deployment to resource-constrained platforms, e.g., mobile phones and small embedded systems (Baykal et al., 2022). To alleviate these issues, a vast amount of recent work in machine learning has focused on methods to generate compact, powerful network models without the need for massive labeled data sets.

Knowledge Distillation (KD) (Buciluǎ et al., 2006; Hinton et al., 2015; Gou et al., 2021; Beyer et al., 2021) is a general purpose approach that has shown promise in generating lightweight powerful models even when a limited amount of labeled data is available (Chen et al., 2020). The key idea is to use a large teacher model trained on labeled examples to train a compact student model so that its predictions imitate those of the teacher. The premise is that even a small student is capable enough to *represent* complicated solutions, even though it may lack the inductive biases to appropriately learn representations from limited data on its own (Stanton et al., 2021; Menon et al., 2020). In practice, KD often leads to significantly more predictive models than otherwise possible with training in isolation (Chen et al., 2020; Xie et al., 2020; Gou et al., 2021; Cho & Hariharan, 2019).

Knowledge Distillation has recently been used to obtain state-of-the-art results in the semi-supervised setting where a small number of labeled and a large number of unlabeled examples are available (Chen et al., 2020; Pham et al., 2021; Xie et al., 2020). *Semi-supervised KD* entails training a teacher model on the labeled set and using its soft labels on the unlabeled data to train the student. The teacher is often a pre-trained model and may also be a generic large model such as GPT-3 (Brown et al., 2020)

or PaLM (Chowdhery et al., 2022). The premise is that a large teacher model can more aptly extract knowledge and learn from a labeled data set, which can subsequently be distilled into a small student.

Despite its widespread success, KD generally suffers from various degrees of *confirmation bias* and *inefficiency* in modern applications to semi-supervised learning. Confirmation bias (Pham et al., 2021; Liu & Tan, 2021; Arazo et al., 2020; Beyer et al., 2021) is the phenomenon where the student exhibits poor performance due to training on noisy or inaccurate teacher soft-labels. Here, inaccuracy refers to the inconsistency between the teacher's predictions for the unlabeled inputs and their groundtruth labels. Feeding the student inaccurate soft-labels leads to increased confidence in incorrect predictions, which consequently produces a model that tends to resist new changes and perform poorly overall (Liu & Tan, 2021; Arazo et al., 2020). At the same time, large-scale applications often require the teacher's predictions for billions of unlabeled points. For instance, consider distilling knowledge from GPT-3 to train a powerful student model. As of this writing, OpenAI charges 6c per 1k token predictions (OpenAI, 2022). Assuming just 1M examples to label and an average of 100 tokens per example leads to a total cost of $6M. Hence, it is highly desirable to acquire the most helpful – i.e., informative and correct – soft-labels subject to a labeling budget (GPT-3 API calls) to obtain the most powerful student model for the target application.

Thus, it has become increasingly important to develop KD methods that are both query-efficient and robust to labeling inaccuracies. Prior work in this realm is limited to tackling *either* distillation efficiency (Liang et al., 2022; Xu et al., 2020), by combining mix-up (Zhang et al., 2017) and uncertainty-based sampling (Roth & Small, 2006), or robustness (Pham et al., 2021; Liu & Tan, 2021; Arazo et al., 2020; Zheng et al., 2021; Zhang et al., 2020), through clever training and weighting strategies, but *not both of these objectives at the same time*. In this paper, we present a simple-to-implement method that finds a sweet spot and improves over standard techniques. Relatedly, there has been prior work in learning under label noise (see Song et al. (2022) for a survey), however, these works generally assume that the noisy labels are available (i.e., no active learning component) or impose assumptions on the type of label noise (Younesian et al., 2021). In contrast, we assume that the label noise can be fully adversarial and that we do not have full access to even the noisy labels.

To the best of our knowledge, this work is the first to consider the problem of importance sampling for simultaneous efficiency and robustness in knowledge distillation. To bridge this research gap, we present an efficient algorithm with provable guarantees to identify unlabeled points with soft-labels that tend to be simultaneously informative and accurate. Our approach is parameter-free, imposes no assumptions on the problem setting, and can be widely applied to any network architecture and data set. At its core lies the formulation of an optimization problem that simultaneously captures the objectives of efficiency and robustness in an appropriate way. In particular, this paper contributes:

1. A mathematical problem formulation that captures the joint objective of training on *informative* soft-labels that are accurately labeled by the teacher in a query-efficient way
2. A near linear time, parameter-free algorithm to optimally solve it
3. Empirical results on benchmark data sets and architectures with varying configurations that demonstrate the improved effectiveness of our approach relative to the state-of-the-art
4. Extensive empirical evaluations that support the widespread applicability and robustness of our approach to varying scenarios and practitioner-imposed constraints.

## 2 PROBLEM STATEMENT

We consider the semi-supervised classification setting where we are given a small labeled set $\mathcal{X}_L$ – typically tens or hundreds of thousands of examples – together with a large unlabeled set $\mathcal{X}_U$, typically on the order of millions or billions. The goal is to leverage both the labeled and unlabeled sets to efficiently and reliably train a compact, powerful model $\theta_{\text{student}}$. To do so, we use knowledge distillation (Xie et al., 2020; Liang et al., 2020) where the labeled points are used to train a larger, (often pre-trained) teacher model that can then be used to educate a small model (the *student*). We emphasize that the teacher may be a pre-trained model, however, it is not trained on the unlabeled set $\mathcal{X}_U$. The distillation process entails using the soft-labels of the teacher for the unlabeled points. The student is then trained on these soft-labeled points along with the original labeled data set. The key insight is that the large, pre-trained teacher model can more aptly learn representations from the limited data, which can then be imitated by the student.

Somewhat more formally, we are given two input sets $\mathcal{X}_L, \mathcal{X}_U$ independently and randomly drawn from the input space $\mathcal{X} \subseteq \mathbb{R}^d$. We assume that we have access to the hard labels $\mathcal{Y}_L \in \{0,1\}^k$ for the instances in $\mathcal{X}_L$, but not those in $\mathcal{X}_U$ and that $|\mathcal{X}_L| \ll |\mathcal{X}_U|$. Consistent with modern ML applications, we assume that a validation data set of labeled points is available. We will use a slight abuse in notation and refer to the set of labeled data points as $(\mathcal{X}_L, \mathcal{Y}_L)$ to denote the set of $(x, y)$ labeled pairs. We assume large-scale applications of KD where the teacher is exceedingly large to the extent that querying the teacher soft-label $f_{\theta_{\text{teacher}}}(x) \in [0,1]^k$ for an unlabeled point $x \in \mathcal{X}_U$ is costly and soft-labeling all of $\mathcal{X}_U$ is infeasible. In the following, we introduce and motivate robust active distillation to conduct this process efficiently and reliably.

## 2.1 ACTIVE DISTILLATION

The objective of active distillation is to query the minimum number of teacher soft-labels for points in $\mathcal{X}_U$ in order to train a high-performing student model $\theta_{\text{student}}$ in a computationally and financially-efficient way. This process is shown in Alg. 1. Here, we conduct $T$ active distillation iterations after training the student and the teacher models on the training set $\mathcal{P}$, which initially only includes the set of hard-labeled points. On Line 8 and throughout, $f_\theta(x) \in [0,1]^k$ denotes the softmax output of a neural network model $\theta$ with respect to input $x$. At each iteration (Lines 5-10, Alg. 1), we use a given querying algorithm, SELECT, to identify the most helpful $b$ unlabeled points to soft-label by the teacher based on the most up-to-date student model $\theta_{t-1}$ (Line 6). The selected points are then soft-labeled by the teacher and added to the (expanding) training set $\mathcal{P}$. Subsequently, the student is trained using the Kullback-Leibler (KL) Divergence (Hinton et al., 2015) as the loss function on the training set $\mathcal{P}$ which includes both the hard-labeled points $(\mathcal{X}_L, \mathcal{Y}_L)$ and the accumulated soft-labeled ones. We follow the standard convention in active learning (Ren et al., 2021) and efficient distillation (Liang et al., 2020; Xu et al., 2020) and train the student model from scratch on Line 9.

---

**Algorithm 1** ACTIVEDISTILLATION

**Input:** a set of labeled points $(\mathcal{X}_L, \mathcal{Y}_L)$, a set of unlabeled points $\mathcal{X}_U$, the number of points to soft-label per iteration $b \in \mathbb{N}_+$, and a selection algorithm $\text{SELECT}(\bar{\mathcal{X}}, \theta, b)$ that selects a sample of size $b$ from $\bar{\mathcal{X}}$

1: $\mathcal{P} \leftarrow (\mathcal{X}_L, \mathcal{Y}_L)$; {Training set thus far; initially only hard-labeled points}
2: $\theta_{\text{teacher}} \leftarrow \text{TRAIN}(\mathcal{P}, \theta_{\text{teacher}}^{\text{random}})$; {Train teacher on labeled data starting from random initialization}
3: $\theta_0 \leftarrow \text{TRAIN}(\mathcal{P}, \theta_{\text{student}}^{\text{random}})$; {Train student on labeled data starting from random initialization}
4: $\mathcal{S} \leftarrow \emptyset$; {Set of inputs that have been soft-labeled}
5: **for** $t \in \{1, \ldots, T\}$ **do**
6: $\quad \mathcal{S}_t \leftarrow \text{SELECT}(\mathcal{X}_U \setminus \mathcal{S}, \theta_{t-1}, b)$ {Select $b$ points to be soft-labeled by $\theta_{\text{teacher}}$}
7: $\quad \mathcal{S} \leftarrow \mathcal{S} \cup \mathcal{S}_t$ {Add new points so we do not sample them again}
8: $\quad \mathcal{P} \leftarrow \mathcal{P} \cup \{(x, f_{\theta_{\text{teacher}}}(x)) : x \in \mathcal{S}_t\}$ {Soft-label points and add them to the training set}
9: $\quad \theta_t \leftarrow \text{TRAIN}(\mathcal{P}, \theta_{\text{student}}^{\text{random}})$ {Train network with the additional soft-labeled points from scratch}
10: **end for**
11: **return** $\theta_T$

---

The active distillation problem is deeply related to the problem of *active learning*, where the objective is to query the labels of only the most informative points in order to minimize labeling costs. To this end, prior approaches in efficient KD (Xu et al., 2020; Liang et al., 2020) have proposed methods inspired by margin-based sampling (Balcan et al., 2007; Roth & Small, 2006), a popular and widely used active learning algorithm (Ren et al., 2021). Margin-based sampling is one example of uncertainty-based sampling, other examples are clustering-based selection (Sener & Savarese, 2017; Ash et al., 2019), model uncertainty (Gal et al., 2017), and adversarial proximity (Ducoffe & Precioso, 2018) (see (Ren et al., 2021) for a survey). In the following, we consider margin-based sampling due to its simplicity and prior application to efficient distillation by related work (Liang et al., 2020; Xu et al., 2020). Margin-based sampling for KD is an intuitive and simple-to-implement idea where the teacher predictions for inputs that the student is most uncertain about are queried. For an input $x$ and prediction $i^* = \arg\max_{i \in [k]} f_{\theta_{\text{student}}}(x)_i$, the uncertainty is measured in terms of the *margin* between the top-2 highest probability entries, i.e., $\text{margin}(x) = f_{\theta_{\text{student}}}(x)_{i^*} - \max_{i \in [k] \setminus i^*} f_{\theta_{\text{student}}}(x)_i$.

## 2.2 RESEARCH GAP

Despite the widespread success of margin-based sampling in active learning, we claim that it is generally ill-suited for knowledge distillation due to its tendency to amplify confirmation bias, leading

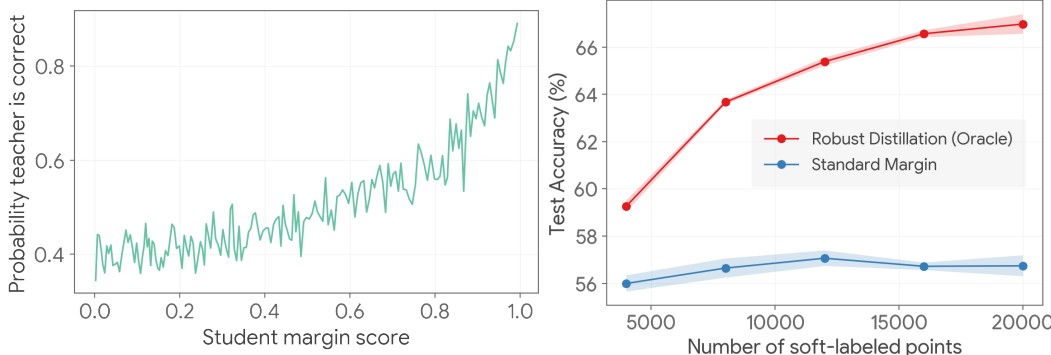

Figure 1: **Left:** teacher's accuracy (overall $60.7\%$) relative to the student's (overall $49.4\%$) *margin score*; points with lower margin tend to be incorrectly classified by the teacher. Plot was generated by averaging teacher's accuracy over 100 closest-margin points. **Right:** Performance of robust distillation (red), which picks the lowest-margin points *among those correctly labeled by the teacher*, compared to that of margin (blue).

to poor student performance. To observe this, note that the objective of margin-based sampling – and more generally, other uncertainty-based query methods – is to query the soft-labels of inputs for which the student is most uncertain about ("hard" instances). However, hard instances for the student are often hard to predict correctly by the teacher. Hence, the soft-labels for these points are more likely to be incorrect with respect to the groundtruth labels, leading to misleading student training.

Fig. 1 shows an instance of this phenomenon for CIFAR10 with ResNet student-teacher architectures of varying depth. As the figure depicts, the teacher tends to predict incorrect labels for points with low student margin (hard instances), and conversely, tends to be highly accurate on points with high margin (easy instances). This suggests that there is an inherent trade-off between efficiency (minimizing queries) and robustness (mitigating confirmation bias) that needs to be considered. That is, we would like to pick informative points for the student for efficiency, but these informative points tend to be incorrectly classified by the student which leads to misguided training and poor performance. Is it possible to simultaneously achieve both in a principled way? We label this problem *Robust Active Distillation* and propose a method to solve it in the following section.

## 3 ROBUST ACTIVE DISTILLATION (RAD)

### 3.1 BACKGROUND

The margin algorithm (Liang et al., 2020; Roth & Small, 2006) selects the $b$ points with the lowest margin scores $\mathrm{margin}(x)$, where $b$ is our soft-label budget. Let $\mathrm{margin}_i$ be shorthand for the margin of each unlabeled input $\mathrm{margin}(x_i)$ and observe that its *gain* or informativeness can be quantified as $g_i = 1 - \mathrm{margin}_i$. Given a budget $b$, note that the margin-sampling algorithm corresponds to the optimal solution of the following optimization problem where the objective is to generate a probability distribution that maximizes the expected sum of gains,

$$\max_{p \in \Delta_b} \mathbb{E}_{\mathcal{S} \sim p} \left[ \sum_{i \in \mathcal{S}} g_i \right], \text{ where } \Delta_b = \{p \in [0,1]^n : \sum_{i \in [n]} p_i = b\}. \tag{1}$$

As discussed previously, this formulation solely focuses on the informativeness of the points and does not consider the increased likelihood of mislabeling by the teacher.

**Robust Distillation** To extend (1) so that it is robust to possible teacher mislabeling, consider the masks $c_i = \mathbb{1}\{\text{teacher labels point i correctly}\}$ for each $i \in [n]$ where $\mathbb{1}\{x\} = 1$ if $x$ is true and 0 otherwise. Equipped with this additional variable, one way to explicitly mitigate confirmation bias and simultaneously pick informative samples is to reward points that are correctly labeled by the teacher by assigning gains as before, but penalize those that are incorrectly labeled via losses. This can be done by using the modified gains in the context of (1)

$$\tilde{g}_i = g_i c_i - (1 - c_i)\ell_i = \begin{cases} g_i, & \text{if teacher labels point i correctly} \\ -\ell_i, & \text{otherwise} \end{cases} \quad \forall i \in [n].$$

In words, this means that if the point $i$ is correctly labeled by the teacher we assign the standard margin-based gain $g_i = (1 - \text{margin}_i)$ as before; otherwise, we penalize the selection by assigning $-\ell_i$ for some loss $\ell_i \geq 0$. This leads to the following general problem of robust distillation

$$\max_{p \in \Delta_b} \mathbb{E}_{i \sim p} \left[ g_i c_i - (1 - c_i) \ell_i \right]. \tag{2}$$

The optimal solution to problem (2) corresponds to picking the $b$ most informative (highest gain) points *among those that are predicted correctly by the teacher*, i.e., those points $i \in [n]$ with highest $g_i$ subject to $c_i = 1$. This approach is shown as Robust Distillation (Oracle) in Fig. 1 (right). Fig. 1 exemplifies the effect of inaccurate examples on student training (see also (Pham et al., 2021)). If we had knowledge of $(c_i)_{i \in [n]}$, then we could optimally solve (2) to obtain significant improvements over the standard margin algorithm. Unfortunately, perfect knowledge of whether the teacher labels each point correctly or not, i.e., $(c_i)_{i \in [n]}$, is not possible in the semi-supervised setting.

## 3.2 OUR APPROACH

We consider a general and robust approach that simultaneously leverages instance-specific knowledge without having to know the masks $(c_i)_{i \in [n]}$ individually. Suppose that we only know that the teacher mislabels $m$ points out of the $n$ unlabeled inputs $\mathcal{X}_B$ instead. Can we generate a sampling distribution so that no matter which of the $m$ points are labeled incorrectly by the teacher, our expected gain is high? Assuming $b = 1$ for simplicity, we arrive at the extension of the formulation in (2)

$$\max_{p \in \Delta_1} \min_{c \in \Delta_{n-m}} \mathbb{E}_{i \sim p} \left[ g_i c_i - (1 - c_i) \ell_i \right], \text{ where } \Delta_k = \{ q \in [0,1]^n : \sum_{i \in [n]} q_i = k \}. \tag{3}$$

Problem (3) has the following game theoretic interpretation. We go first and pick a sampling distribution $p$ over the points. In response, an adversary decides which points are misclassified (i.e, $c_i = 0$) by the teacher subject to the constraint that it can set $c_i = 0$ for at most $m \leq n$ of them since $c \in \Delta_{n-m}$. Given the linear structure of the problem, it turns out that we can invoke von Neumann's Minimax Theorem (Neumann, 1928) which states that the equilibrium point is the same regardless of whether we go first and pick the probability distribution $p \in \Delta_1$ or the adversary goes first and picks $(c_i)_{i \in [n]}$. By exploiting this connection, we obtain a closed form solution as formalized below.

**Theorem 1.** *Suppose* $g_1 \geq g_2 \geq \cdots \geq g_n > 0$, *and define* $G_k = \sum_{i \leq k} g_i / (g_i + \ell_i)$ *and* $H_k = \sum_{i \leq k} 1 / (g_i + \ell_i)$. *For* $G_n \geq m$, *an optimal solution* $p^* \in \Delta_1$ *to* (3) *is given by*

$$p_i^* = \frac{1}{H_{k^*}(g_i + \ell_i)} \text{ if } i \leq k^* \text{ and } p_i^* = 0 \text{ otherwise, where } k^* = \text{argmax}_{k \in [n]} \frac{G_k - m}{H_k}.$$

*The distribution can be computed in linear time (assuming sorted $g$) and achieves an objective value of* $\text{OPT}(k^*) := \frac{G_{k^*} - m}{H_{k^*}}$.

We sketch the proof here. The full proof can be found in the Appendix (Sec. C).

*Proof sketch.* Let

$$\text{OBJ}(p, c) = \sum_{i \in [n]} p_i (g_i c_i - (1 - c_i) \ell_i).$$

Substituting $p^*$ from Thm. 1 and considering a minimizing value for $c \in \Delta_{n-m}$, it is possible to show that $p^* \in \Delta_1$ and

$$\text{OPT}(k^*) = \min_{c \in \Delta_{n-m}} \text{OBJ}(p^*, c) \leq \max_{p \in \Delta_1} \min_{c \in \Delta_{n-m}} \text{OBJ}(p, c).$$

On the other hand, let $c_i^* = \min\left(1, \frac{\text{OPT}(k^*) + \ell_i}{g_i + \ell_i}\right)$. With a little more work, using the fact that $g_{k^*} \geq \text{OPT}(k^*) \geq g_{k^*+1}$, we can show similarly that $c^* \in \Delta_{n-m}$ and

$$\text{OPT}(k^*) = \max_{p \in \Delta_1} \text{OBJ}(p, c^*) \geq \min_{c \in \Delta_{n-m}} \max_{p \in \Delta_1} \text{OBJ}(p, c).$$

With these inequalities in hand, we apply the Minimax Theorem (Neumann, 1928), which yields

$$\text{OPT}(k^*) \leq \max_{p \in \Delta_1} \min_{c \in \Delta_{n-m}} \text{OBJ}(p, c) = \min_{c \in \Delta_{n-m}} \max_{p \in \Delta_1} \text{OBJ}(p, c) \leq \text{OPT}(k^*).$$

Hence, $p^*$ does indeed obtain the optimal value, $\text{OPT}(k^*)$. $\square$

**RAD Loss** Equipped with Theorem 1, all that remains is to specify the losses in (3). Prior work on confirmation bias has shown that even a small number misguided soft-labels can derail the student's performance and significantly impact its predictive capability (Liu & Tan, 2021). Additionally, the harm of an incorrectly labeled point may be even more pronounced when the student is uncertain about that point. To model this, we consider instantiating our general problem formulation (3) with losses that are *relative to the gain* $\ell_i = -wg_i$ for each $i \in [n]$ where $w \in [0, 1]$ is a weight parameter that controls the magnitude of the penalization. This formulation purposefully leads to higher penalties for misclassified points that the student is already unsure about (high gain) to mitigate confirmation bias, and leads to the following optimization problem which is the focus of this paper

$$\max_{p \in \Delta_1} \min_{c \in \Delta_{n-m}} \mathbb{E}_{i \sim p} \left[ g_i c_i - (1 - c_i) w g_i \right]. \tag{4}$$

Invoking Thm. 1 with the relative gain losses as described above immediately leads to the following, which, along with the choice of $w$ below, describes the algorithm RAD that we propose in this paper.

**Corollary 1.** *An optimal solution $p^* \in \Delta_1$ to (4) has $k$ non-zero entries $\mathcal{I}_{k^*} \subseteq [n]$ corresponding to the $k^*$ indices of the largest entries of g, with*

$$p_i^* = \frac{1}{g_i \sum_{j \in \mathcal{I}_{k^*}} g_j^{-1}} \quad \forall i \in \mathcal{I}_{k^*} \quad where \quad k^* = \text{argmax}_{k \in [n]} \frac{k - (1+w)m}{\sum_{j \in \mathcal{I}_k} g_j^{-1}}.$$

*The distribution $p^*$ can be computed in $\Theta(n \log n)$ time, with $\text{OPT}(k^*) = {(k^* - (1+w)m)}/{\sum_{j \in \mathcal{I}_{k^*}} g_j^{-1}}$.*

**Choice of $w$** Although RAD can be applied with any user-specified choice of $w$, we use the theoretically-motivated weight of $w = (1 - m/n)$ as the relative penalization constant in our experiments. This choice of $w$ guarantees that the optimal value (expected gain) of (4) (see Corollary (1)) is non-negative — if the expected gain were negative, we would be better off not sampling at all. *This default value for $w$ makes* RAD *parameter-free.* Extensive empirical evaluations with varying values of $w$ are presented in Sec. D.5 of the Appendix.

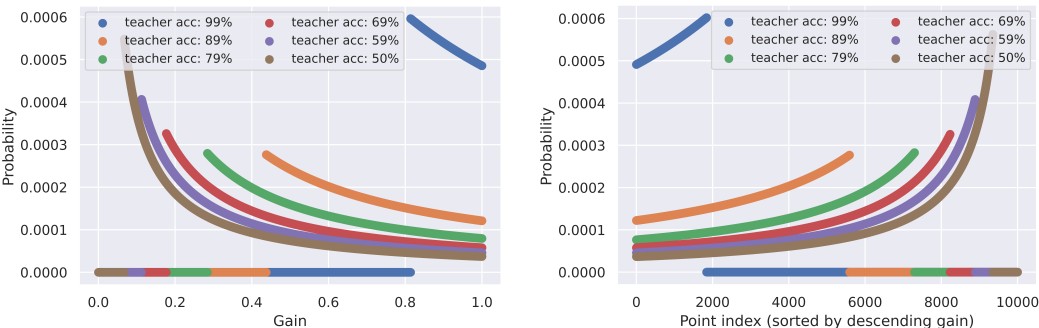

Figure 2: The optimal solution to (4) given by Corollary 1 for various values for teacher accuracy (varying $m$). Even when most of the samples are labeled correctly by the teacher, our method is purposefully cautious and does not allocate all of the probability (of sampling) mass on the highest gain points.

We observe several favorable properties of RAD's sampling distribution in Fig. 2, which depicts the computed distribution on a synthetic scenario with gains drawn uniformly at random from $[0, 1]$. For one, the sampling distribution tends to allocate less probability mass to the highest gain items. As prior work has shown, this is desirable because the hardest (highest gain) examples tend to be outliers or points with noisy labels (Mindermann et al., 2022; Ren et al., 2018). In fact, robust learning approaches typically downweight hard examples for this reason (Kumar et al., 2010), analogous to RAD's sampling behavior. At the same time, Paul et al. (2021) show that the easiest (lowest gain) examples tend to be truly uninformative and the best strategy is to ignore a certain fraction of the highest and lowest gain points. This strategy parallels RAD's computed distribution, where a number of low gain points are ignored and the probability peaks around a region inbetween (see Fig. 2). A prominent benefit of RAD is that this region is computed in a fully automated way as a function of the teacher's accuracy (i.e., amount of label noise). If the teacher is highly accurate, the distribution accordingly concentrates on the highest gain points (blue, Fig. 2); otherwise, it spreads out the probabilities over a larger portion of the points and purposefully assigns lower sampling probability to the highest gain points which are likely to be noisy (e.g., brown, Fig. 2).

### 3.3 IMPLEMENTATION DETAILS

We conclude this section by outlining the practical details of RAD. We follow the setting of this section and set $g_i = 1 - \text{margin}_i$ to define the gains of each point (see Sec. D.6 of the Appendix for evaluations with a differing gain definition). In practice, we use the distribution $q = \min\{bp, 1\}$ when sampling a set of $b$ points, where $p$ is from Corollary 1. This is optimal as long as the probabilities from Corollary 1 (or more generally, Theorem 1) are not heavily concentrated on a few points (i.e., $\max_{i \in [n]} p_i \leq 1/b$). As exemplified in Fig. 2 and experimentally verified in Sec. 4 in all of the evaluated scenarios, we found this to virtually always be the case. Alternatively, the acquisition size $b$ can be adjusted after the single-sample probability distribution is computed so that $b \leq \min_{i \in [n]} 1/p_i$. Since the number of mistakes the teacher makes on $\mathcal{X}_B$, $m$, is not known to us in the semi-supervised setting, we approximate this quantity by first taking the sample mean inaccuracy of a small uniform random sample of $b_{\text{uniform}}$ (see Appendix, Sec. D.1) points as a way to approximate $m$ and bootstrap our approach. We then use our approach with $b' = b - b_{\text{uniform}}$ as described above. By Bernstein's inequality (Bernstein, 1924), this weighted estimate tightly concentrates around the mean, which in turn implies a high-quality approximation of $m$. We theoretically analyze the effect of an approximate $m$ on the quality of the optimal solution in Sec. C of the Appendix (see Lemmas 4 and 5).

## 4 RESULTS

We apply our sample selection algorithm, RAD, to benchmark vision data sets and evaluate its performance in generating high-performance student models on a diverse set of knowledge distillation scenarios. We compare the performance of RAD to the following: (i) MARGIN (Balcan et al., 2007; Roth & Small, 2006) as described in Sec. 3, (ii) UNIFORM, (iii) CLUSTER MARGIN (labeled CM), a state-of-the-art active learning technique (Citovsky et al., 2021), (iv) CORESET, a popular clustering-based active learning algorithm (Sener & Savarese, 2017), (v) ENTROPY, a greedy approach that picks the points with highest student prediction entropy (Holub et al., 2008), and (vi) UNIXKD (Xu et al., 2020), a state-of-the-art active distillation approach based on mix-up (Zhang et al., 2017). We implemented all algorithms in Python and used the TensorFlow (Abadi et al., 2015) deep learning library. We used the hyperparameters specified in the respective papers for all of the compared approaches. For RAD, we use the theoretically-derived setting of $w = 1 - m/n$ as specified in Sec. 3 and emphasize that this makes RAD fully parameter-free.

In Sec. D of the Appendix, we present: the full set of hyper-parameters and experimental details (Sec. D.1); additional evaluations that report statistics beyond test accuracy (Sec. D.3); applications of RAD to the standard active learning setting and comparisons to SOTA approaches (Sec. D.4); experiments with varying $w$ and gain definition to evaluate the robustness of RAD (Sec. D.5 and D.6, respectively); comparisons on a diverse set of knowledge distillation configurations (Sec. D.7). Overall, our empirical evaluations show that RAD uniformly improves on state-of-the-art baselines and demonstrate its off-the-shelf effectiveness without the need to tune or change any hyperparameters.

### 4.1 CIFAR10, CIFAR100, & SVHN

**Setup**   We use ResNet (He et al., 2015), ResNetV2-$\{11, 20, 29\}$ (He et al., 2016), or MobileNet (Howard et al., 2017) with a depth multiplier of 1 as the student and ResNet-50, ResNetV2-$\{56, 110\}$, or MobileNet with a depth multiplier of 2 as the teacher model. We considered the CIFAR10/CIFAR100 (Krizhevsky et al., 2009), SVHN (Netzer et al., 2011), and ImageNet (Deng et al., 2009) data sets. Unless otherwise specified, we use the Adam optimizer (Kingma & Ba, 2014) with a batch size of 128 with data set specific learning rate schedules. We follow the active distillation setting shown in Alg. 1 with various configurations. We use 64 Cloud TPU v4s each with two cores. The full set of hyper-parameters and experimental details can be found in Sec. D of the Appendix.

**Configurations**   We experimented with a diverse set of configurations for the knowledge distillation task. We reported the specific configuration for each plot as part of the plot's title (e.g., see Fig. 3). In context of the variables in the plot heading, we varied the number of epochs that the student is trained for (denoted as $e$), the size of the initial set of labeled points ($|A|$), the number of soft-labels to query per iteration ($b$), and the teacher model ($t$); *resnet* in the configuration refers to ResNetV2-20 as the student and ResNet-50 as the teacher unless otherwise specified. All results were averaged over 10

trials unless otherwise stated. For each trial, we reshuffled the entire data set and picked a random portion (of size $|A|$) to be the labeled data set $\mathcal{X}_L$, and considered the rest to be the unlabeled set $\mathcal{X}_U$.

Figure 3: Evaluations of RAD, state-of-the-art active learning, and active distillation strategies on a diverse set of distillation configurations with varying data sets and network architectures. RAD consistently outperforms competing approaches. Shaded regions correspond to values within one standard deviation of the mean.

In the first set of experiments, we evaluate the effectiveness of each method in generating high-accuracy student models subject to a soft-labeling budget on CIFAR10, CIFAR100, and SVHN data sets with ResNet(v2) and MobileNet architectures of varying sizes. CIFAR10 contains $50,000$ images of size $32 \times 32$ with 10 categories, CIFAR100 has $50,000$ $32 \times 32$ images with 1000 labels, and SVHN consists of $73,257$ real-world images ($32 \times 32$) taken from Google Street View. Fig. 3 depicts the results of our evaluations on a diverse set of knowledge distillation scenarios with varying configurations. We observe a consistent and marked improvement in the student model's predictive performance when our approach is used to actively select the points to be soft-labeled by the teacher. This improvement is often present from the first labeling iteration and persists continuously over the active distillation iterations.

We observe that RAD performs particularly well relative to baselines regardless of the teacher's accuracy. For instance, we see significant improvements with RAD when distilling from a MobileNet teacher on CIFAR100, which has relatively low accuracy (see corresponding plots in Fig. 3). This

observation suggests that the explicit consideration of possible teacher inaccuracy is indeed helpful when distilling from a teacher that may be prone to making mistakes. At the same time, we observe that RAD outperforms state-of-the-art active learning algorithms such as Cluster Margin (CM) and others (MARGIN, ENTROPY) – which do not explicitly consider label noise in the form of incorrect teacher soft-labels – even in instances where the teacher accuracy is as high as $\approx 92\%$ (see SVHN plots in Fig. 3). These observations support RAD's ability to automatically adapt its sampling distribution to the applied scenario based on the approximated teacher inaccuracy.

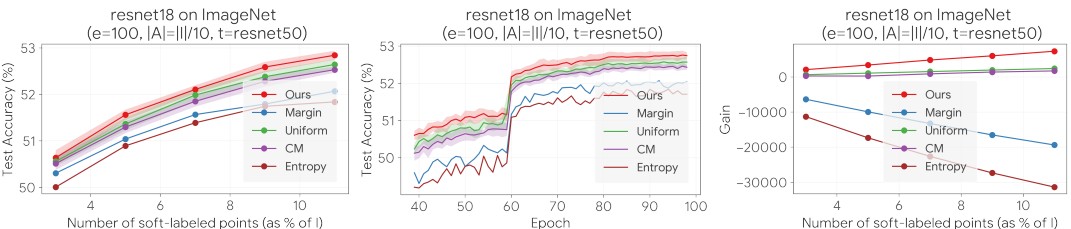

Figure 4: The classification accuracy and gain on the ImageNet data set with ResNets.

## 4.2 IMAGENET EXPERIMENTS

Here, we report the results of our evaluations with ResNet architectures trained on the ImageNet data set $I$, which contains nearly 1.3 million images spanning 1000 categories (Deng et al., 2009). We exclude the UNIXKD and CORESET methods due to resource constraints and the fact that CM supersedes CORESET (Citovsky et al., 2021) and UNIXKD consistently performed poorly on the configurations in the previous subsection. This highlights an additional advantage of our approach: it merely requires a sort ($\mathcal{O}(n \log n)$ total time). This is in contrast to computation- and memory-expensive methods like CORESET and CM that require clustering (see Sec. D.2 for details). Fig. 4 depicts the results of our evaluations, where our method consistently improves upon the compared approaches across all training epochs. Here the teacher is a ResNet50 model and the student is a ResNet18 model. The teacher is trained on the initial labeled dataset $A$ with $|A| = 10\%|I|$. For each choice of the budget $b \in \{3\%|I|, 5\%|I|, 7\%|I|, 9\%|I|, 11\%|I|\}$, we run 3 trials. In the rightmost plot we observe that the gain of our approach (w.r.t. $g_i = 1 - \text{margin}_i$) is higher than that of the competing approaches, and that the gain correlates well with the test accuracy of the student, which reaffirms the practical validity of our formulation (Sec. 3).

## 5 CONCLUSION

In this paper, we considered the problem of efficient and robust knowledge distillation in the semi-supervised learning setting with a limited amount of labeled data and a large amount of unlabeled data. We formulated the problem of robust active distillation and presented a near linear-time algorithm with provable guarantees to solve it optimally. To the best of our knowledge, our work is the first to consider importance sampling for informative and correctly labeled soft-labels to enable efficiency and robustness in large-scale knowledge distillation tasks. Our method is parameter-free and simple-to-implement. Our experiments on popular benchmark data sets with a diverse set of configurations showed a consistent and notable improvement in the test accuracy of the generated student model relative to those generated by state-of-the-art methods.

**Limitations and future work** In future work, we plan to establish a deeper theoretical understanding on the trade-offs of the various instantiations of our general framework, (3) in Sec. 3, on the *test accuracy* of the student model. For example, it is not clear whether defining the gain as $g_i = 1 - \text{margin}_i$ or $g_i = \exp(-\text{margin}_i)$ is more appropriate, even though both definitions lead to gains that are monotonically increasing with the uncertainty of the student. Besides considering teacher accuracy in the robust formulation, we plan to also consider other relevant metrics such as student-teacher disagreement to construct more informed distributions. Overall, we envision that our approach can be used in high-impact applications to generate powerful student models by efficiently distilling the knowledge of large teachers in the face of limited labeled data.

## REPRODUCIBILITY STATEMENT

Our algorithm is fully-specified (Corollary 1), simple-to-implement, and parameter-free (Sec. 3). We provide the full details and hyperparameters required to reproduce our results in Sec. 4 and Sec. D.1 of the Appendix. We specify descriptions of how the competing algorithms were implemented, including the hyperparameter settings. We provide precise theoretical results (Sec. 3 in the main body and Sec. C of the Appendix) that clearly specify the assumptions and provide full proofs and additional helper lemmas in the Appendix (Sec. C). Our evaluations use publicly available and easily accessible data sets and models.

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

## A    APPENDIX

In this supplementary material, we provide the details of batch sampling (in Sec. B), full proofs of the results in Sec. 3 and additional theoretical results (in Sec. C), and details of experiments in Sec. 4 and additional evaluations (in Sec. D).

## B    IMPLEMENTATION DETAILS

To supplement our discussion in Sec. 3, we provide additional details regarding the batch sampling procedure. One approach is to iterate over the points $i \in [n]$ and pick each point with probability $q_i$. If $\sum_{i \in [n]} q_i = b$, then this procedure samples $b$ points in expectation. A more principled approach is to use randomized dependent rounding (Chekuri et al., 2009) which samples *exactly* $b$ points given a distribution that sums to $b$. This procedure is shown as Alg. 2, and an efficient implementation of it runs in $\mathcal{O}(n)$ time (Chekuri et al., 2009).

---

**Algorithm 2** DEPROUND

---

**Inputs**: Probabilities $p \in [0, 1]^n$ such that $\sum_{i \in [n]} p_i = b$
**Output:** set of indices $\mathcal{I} \subset [n]$ with $|\mathcal{I}| = b$

1: **while** $\exists i \in [n]$ such that $0 < p_i < 1$ **do**
2:    Pick $i, j \in [n]$ with $i \neq j$, $0 < p_i < 1$, and $0 < p_j < 1$
3:    $\alpha \leftarrow \min(1 - p_i, p_j)$
4:    $\beta \leftarrow \min(p_i, 1 - p_j)$
5:    Update $p_i$ and $p_j$
$$(p_i, p_j) = \begin{cases} (p_i + \alpha, p_j - \alpha) & \text{with probability } \frac{\beta}{\alpha + \beta}, \\ (p_i - \beta, p_j + \beta) & \text{with probability } 1 - \frac{\beta}{\alpha + \beta}. \end{cases}$$
6: **end while**
7: $\mathcal{I} \leftarrow \{i \in [n] : p_i = 1\}$
   **return** $\mathcal{I}$

---

## C    PROOFS & ADDITIONAL ANALYSIS

### C.1    PROOF OF THEOREM 1

**Theorem 1.** *Suppose* $g_1 \geq g_2 \geq \cdots \geq g_n > 0$, *and define* $G_k = \sum_{i \leq k} g_i / (g_i + \ell_i)$ *and* $H_k = \sum_{i \leq k} 1 / (g_i + \ell_i)$. *For* $G_n \geq m$, *an optimal solution* $p^* \in \Delta_1$ *to* (3) *is given by*

$$p_i^* = \frac{1}{H_{k^*}(g_i + \ell_i)} \text{ if } i \leq k^* \text{ and } p_i^* = 0 \text{ otherwise, where } k^* = \operatorname{argmax}_{k \in [n]} \frac{G_k - m}{H_k}.$$

*The distribution can be computed in linear time (assuming sorted $g$) and achieves an objective value of* $\operatorname{OPT}(k^*) := \frac{G_{k^*} - m}{H_{k^*}}$.

*Proof.* The proof relies on the the Minimax Theorem (Neumann, 1928), which yields

$$\min_{c \in \Delta_{n-m}} \max_{p \in \Delta_1} f(p, c) = \max_{p \in \Delta_1} \min_{c \in \Delta_{n-m}} f(p, c).$$

We will further use two claims, which we'll prove shortly.

**Claim 2.** *Let* $p^*$, $k^*$, *and* $N(k)$ *be defined as in the statement of the lemma. Then* $p^* \in \Delta_1$ *and* $N(k^*) = \min_{c \in \Delta_{n-m}} f(p^*, c)$.

**Claim 3.** *Let* $k^*$ *and* $N(k)$ *be defined as in the statement of the lemma. Further, define*

$$c_i^* = \begin{cases} \frac{N(k^*) + \ell_i}{g_i + \ell_i} & \text{if } i \leq k^* \\ 1 & \text{otherwise} \end{cases}$$

*Then $c^* \in \Delta_{n-m}$ and $N(k^*) = \max_{p \in \Delta_1} f(p, c^*)$.*

Given the claims, we see

$$
\begin{aligned}
N(k^*) = \min_{c \in \Delta_{n-m}} f(p^*, c) &\leq \max_{p \in \Delta_1} \min_{c \in \Delta_{n-m}} f(p, c) \text{ by Claim 2} \\
&= \min_{c \in \Delta_{n-m}} \max_{p \in \Delta_1} f(p, c) \text{ by Minimax} \\
&\leq \max_{p \in \Delta_1} f(p, c^*) = N(k^*) \text{ by Claim 3}
\end{aligned}
$$

That is, $p^*$ attains the maximum, which is $N(k^*)$, as we wanted.

We now prove the claims. Before beginning, we will first simplify $f(p, c)$ somewhat, finding

$$
\begin{aligned}
f(p, c) &= \sum_i g_i p_i c_i - \sum_i \ell_i p_i (1 - c_i) \\
&= \sum_i (c_i(g_i + \ell_i) - \ell_i) p_i
\end{aligned}
$$

*Proof of Claim 2.* We first show $p^* \in \Delta_1$. We have

$$
\sum_{i \leq n} p_i^* = \sum_{i \leq k^*} p_i^* = \sum_{i \leq k^*} \frac{1}{H_{k^*}(g_i + \ell_i)} = \frac{H_{k^*}}{H_{k^*}} = 1
$$

And since $p_i^* \geq 0$ for all $i$, it immediately follows that $p_i^* \leq 1$ as well.

We now show $N(k^*) = \min_{c \in \Delta_{n-m}} f(p^*, c)$. To this end,

$$
\begin{aligned}
f(p^*, c) &= \sum_{i \leq n} (c_i(g_i + \ell_i) - \ell_i) p_i \\
&= \sum_{i \leq k^*} \frac{c_i(g_i + \ell_i) - \ell_i}{H_{k^*}(g_i + \ell_i)} \\
&= \sum_{i \leq k^*} \frac{c_i(g_i + \ell_i)}{H_{k^*}(g_i + \ell_i)} - \sum_{i \leq k^*} \frac{(g_i + \ell_i)}{H_{k^*}(g_i + \ell_i)} + \sum_{i \leq k^*} \frac{g_i}{H_{k^*}(g_i + \ell_i)} \\
&= \frac{1}{H_{k^*}} \sum_{i \leq k^*} c_i - \frac{k^*}{H_{k^*}} + \frac{G_{k^*}}{H_{k^*}}
\end{aligned}
$$

Clearly, for $c^* \in \Delta_{n-m}$, this expression is minimized when $c_i = 1$ for $i > k^*$ and $\sum_{i \leq k^*} c_i = k^* - m$. So we have

$$
\begin{aligned}
\min_{c \in \Delta_{n-m}} f(p^*, c) &= \frac{1}{H_{k^*}}(k^* - m) - \frac{k^*}{H_{k^*}} + \frac{G_{k^*}}{H_{k^*}} \\
&= \frac{G_{k^*} - m}{H_{k^*}} = N(k^*)
\end{aligned}
$$

as claimed. $\square$

We now prove our second claim.

*Proof of Claim 3.* We first show that $N(k^*) \geq g_i$ for $i > k^*$.[1] Observe that if $\frac{a}{b} \geq \frac{a+c}{b+d}$, then $\frac{a}{b} \geq \frac{c}{d}$ for non-negative values (and $b > 0, d > 0$). Notice

$$
\frac{G_{k^*} - m}{H_{k^*}} = N(k^*) \geq N(k^* + 1) = \frac{G_{k^*} + g_{k^*+1}/(g_{k^*+1} + \ell_{k^*+1}) - m}{H_{k^*} + 1/(g_{k^*+1} + \ell_{k^*+1})}
$$

---

[1]In the border case that $k^* = n$, we can add a dummy item with $g_{n+1} = 0$ and the claim follows trivially.

By our observation,

$$N(k^*) \geq \frac{g_{k^*+1}/(g_{k^*+1} + \ell_{k^*+1})}{1/(g_{k^*+1} + \ell_{k^*+1})} = g_{k^*+1} \geq g_i \text{ for } i > k^*.$$

Similarly, we show $N(k^*) \leq g_i$ for $i \leq k^*$, using the observation that if $\frac{a-c}{b-d} \leq \frac{a}{b}$, then $\frac{a}{b} \leq \frac{c}{d}$ for non-negative values (and $b - d > 0$ and $d > 0$). Notice

$$N(k^* - 1) = \frac{G_{k^*} - g_{k^*}/(g_{k^*} + \ell_{k^*}) - m}{H_{k^*} - 1/(g_{k^*} + \ell_{k^*})} \leq \frac{G_{k^*} - m}{H_{k^*}} = N(k^*)$$

And again, by our observation,

$$N(k^*) \leq \frac{g_{k^*}/(g_{k^*} + \ell_{k^*})}{1/(g_{k^*} + \ell_{k^*})} = g_{k^*} \leq g_i \text{ for } i \leq k^*. \tag{5}$$

Now we're ready to prove the claim.

We first show $c^* \in \Delta_{n-m}$. Since $N(k^*) \leq g_i$ for $i \leq k^*$, we see $c_i^* = \frac{N(k^*)+\ell_i}{g_i+\ell_i} \leq 1$ for $i \leq k^*$. So $c_i^* \in [0, 1]$. Further,

$$\begin{aligned}
\sum_{i \leq n} c_i^* &= \sum_{i \leq k^*} \frac{N(k^*) + \ell_i}{g_i + \ell_i} + \sum_{k^* < i \leq n} 1 \\
&= \sum_{i \leq k^*} \frac{N(k^*) - g_i}{g_i + \ell_i} + \sum_{i \leq k^*} \frac{g_i + \ell_i}{g_i + \ell_i} + \sum_{k^* < i \leq n} 1 \\
&= N(k^*)H_{k^*} - G_{k^*} + n \\
&= G_{k^*} - m - G_{k^*} + n = n - m
\end{aligned}$$

That is, $c^* \in \Delta_{n-m}$.

Finally, we show $N(k^*) = \max_{p \in \Delta_1} f(p, c^*)$. Note that $c_i^*(g_i + \ell_i) - \ell_i = N(k^*)$ for $i \leq k^*$, while $c_i^*(g_i + \ell_i) - \ell_i = g_i$ for $i > k^*$. We have

$$\begin{aligned}
f(p, c^*) &= \sum_i (c_i^*(g_i + \ell_i) - \ell_i)p_i \\
&= \sum_{i \leq k^*} N(k^*)p_i + \sum_{i > k^*} g_i p_i
\end{aligned}$$

From above, $N(k^*) \geq g_i$ for $i > k^*$. So the expression is maximized for $p \in \Delta_1$ when $p_i = 0$ for $i > k^*$ and $\sum_{i \leq k^*} p_i = 1$. Hence,

$$\max_{p \in \Delta_1} f(p, c^*) = N(k^*)$$

as we wanted. □

□

## C.2 Effect of approximating m

Here, we prove that an approximately optimal solution can be obtained even if an approximate value of $m$ is used (e.g., via a validation data set). For sake of simplicity, the following lemma considers the case where the losses are 0 in the context of Theorem 1, however, its generalization to general gains and losses – including the relative error formulation that we study in this paper – follows by rescaling the error parameter $\varepsilon$ appropriately. [2] Our main result is that if we have an approximation $\hat{m} \in (1 \pm \varepsilon)m$, then we can use this approximate $m$ to obtain an $(1 - 2\varepsilon m/(2\varepsilon m + (1 + \varepsilon)))$-competitive solution.

---

[2]Note that this result generalizes to the RAD relative loss with $w$ w as discussed in Sec. 3 by considering $m' = (1 + w)m$ (see Corollary 1).

**Lemma 4.** *Let $m \in [n]$ be the groundtruth value of the number of incorrect examples in $\mathcal{X}_B$ and let $k_m$ be the optimal solution with respect to $m$. Assume $g_1 \geq \cdots \geq g_n$ and let $\alpha = g_{k_m}$ and $\beta = \frac{\alpha}{2(2-\alpha)}$. Suppose that we have an approximation $\hat{m}$ of $m$ such that*

$$\hat{m} \in (1 \pm \varepsilon)m \quad \text{with } \varepsilon \in (0, \beta),$$

*then the solution $k_{\tilde{m}}$ with respect to $\tilde{m} = \hat{m}/(1+\varepsilon)$ is $(1 - 2\varepsilon m/(2\varepsilon m + (1+\varepsilon)))$-competitive with the optimal solution, i.e., it satisfies*

$$\text{OBJ}(k_{\tilde{m}}, m) \geq \left(1 - \frac{2\varepsilon m}{2\varepsilon m + (1+\varepsilon)}\right) \text{OPT} = \left(1 - \frac{2\varepsilon m}{2\varepsilon m + (1+\varepsilon)}\right) \text{OBJ}(k_m, m),$$

*where $\text{OBJ}(k, m) = \frac{k-m}{H_k} = \max_{k' \in [n]} \frac{k'-m}{H_{k'}}$.*

*Proof.* For sake of notational brevity, we let $k$ and $k^*$ denote $k_{\tilde{m}}$ and $k_m$, respectively. First, observe that by the assumption of the lemma, we have

$$\tilde{m} = \frac{\hat{m}}{1+\varepsilon} \geq \frac{(1-\varepsilon)m}{1+\varepsilon} \quad \text{and} \quad \tilde{m} \leq m.$$

Note that since $k \geq \tilde{m}$ by the optimality of $k$ with respect to $\tilde{m}$, we have $k \geq \frac{(1-\varepsilon)m}{(1+\varepsilon)(1-\alpha)} > m$ which follows from the optimality condition from the proof of Theorem 1,

$$k \geq g_{k+1}H_k + \tilde{m} \geq \alpha k + \tilde{m} \Rightarrow k \geq \frac{\tilde{m}}{1-\alpha}$$

and $\varepsilon \leq \frac{\alpha}{2(2-\alpha)}$. Since $k$ and $m$ are integral, we have $k \geq m+1 \geq \tilde{m}+1$. Finally, by the optimality of $k$ with respect to $\tilde{m}$, we have

$$\frac{1}{H_k} \geq \frac{k^* - \tilde{m}}{(k - \tilde{m})H_{k^*}}.$$

Putting all of the above together,

$$\begin{aligned}
\text{OBJ}(k, m) &= \frac{k-m}{H_k} \\
&= \frac{k - \tilde{m}}{H_k} - \frac{m - \tilde{m}}{H_k} \\
&\geq \frac{k^* - \tilde{m}}{H_{k^*}} - \frac{(m - \tilde{m})(k^* - \tilde{m})}{(k - \tilde{m})H_{k^*}} \\
&\geq \frac{k^* - m}{H_{k^*}}\left(1 - \frac{m - \tilde{m}}{k - \tilde{m}}\right) \\
&\geq \text{OPT}\left(1 - \frac{m - \tilde{m}}{m - \tilde{m} + 1}\right) \\
&\geq \text{OPT}\left(1 - \frac{2\varepsilon m}{2\varepsilon m + (1+\varepsilon)}\right),
\end{aligned}$$

where the first inequality is by the inequality on the lower bound $1/H_k$ and $k \geq m+1$, the second by $m \geq \tilde{m}$, the third by $k \geq m+1$, and the fourth by

$$m - \tilde{m} \leq \frac{2\varepsilon m}{1+\varepsilon},$$

and rearrangment. $\qquad \square$

### C.3 Approximating $m$

Next, we show how to estimate $m$ using a validation data set $\mathcal{T}$. To do so, we define

$$\text{err}(\mathcal{T}, f_\theta) = \frac{1}{|\mathcal{T}|} \sum_{(x,y) \in \mathcal{T}} \mathbb{1}\{f_\theta(x) \neq y\}$$

where $f_\theta(\cdot) \in [k]$ corresponds to the label prediction of network $\theta$. Note here that

$$m = \text{err}(\mathcal{P}_B, f_\theta)|\mathcal{P}_B|$$

where $\mathcal{P}_B$ corresponds to the points in dataset B.

**Lemma 5.** *For any $\delta \in (0, 1)$, if we use a validation set $\mathcal{T}$ of size $k$ to obtain an approximation $\hat{m} = \text{err}(\mathcal{T}, f_\theta)|\mathcal{P}_B|$ for $m = \text{err}(\mathcal{P}_B, f_\theta)|\mathcal{P}_B|$, then with probability at least $1 - \delta$,*

$$\frac{|m - \hat{m}|}{|\mathcal{P}_B|} \leq \log(4/\delta)\left(\frac{1}{|\mathcal{P}_B|} + \frac{1}{k}\right) + \sqrt{2p(1-p)\log(4/\delta)}\left(\frac{1}{\sqrt{|\mathcal{P}_B|}} + \frac{1}{\sqrt{k}}\right),$$

*where $p = \mathbb{P}_{(x,y)\sim\mathcal{D}}(f_\theta(x) \neq y)$ is the probability of mislabeling for network $f_\theta(\cdot)$.*

*Proof.* Let $\mathcal{T} = \{(x_1, y_1), \ldots, (x_{|\mathcal{T}|}, y_{|\mathcal{T}|})\}$ be a set of $|\mathcal{T}|$ i.i.d. points from the data distribution $\mathcal{D}$ and define $X_i = \mathbb{1}\{f_\theta(x_i) \neq y_i\}$ for each $i \in |\mathcal{T}|$. Letting $X = \frac{1}{|\mathcal{T}|}\sum_i X_i$, we observe that

$$\mathbb{E}[X] = \mathbb{E}_{(x,y)\sim\mathcal{D}}[\mathbb{1}\{f_\theta(x) \neq y\}] = \mathbb{P}_{(x,y)\sim\mathcal{D}}(f_\theta(x) \neq y)$$
$$= p.$$

Since we have a sum of $k = |\mathcal{T}|$ independent random variables each bounded by 1 with variance $\text{Var}(X_i) = p(1-p)$, we invoke Bernstein's inequality ([Bernstein](), [1924]) to obtain

$$\mathbb{P}(|X - p| \geq \varepsilon_\mathcal{T}) \leq 2\exp\left(\frac{-k\varepsilon_\mathcal{T}^2}{2(p(1-p) + \varepsilon_\mathcal{T}/3)}\right),$$

setting the above to $\delta/2$ and solving for $\varepsilon_\mathcal{T}$ yields

$$|X - p| \leq \varepsilon_\mathcal{T} \leq \frac{\log(4/\delta)}{k} + \sqrt{\frac{2p(1-p)\log(4/\delta)}{k}},$$

with probability at least $1 - \delta/2$. Similarly, we can define the random variables $Y_1, \ldots, Y_{|\mathcal{P}_B|}$ so that $Y_i = \mathbb{1}\{f_\theta(x_i) \neq y_i\}$ for each $(x_i, y_i) \in \mathcal{P}_B$. Letting $Y = \frac{1}{|\mathcal{P}_B|}\sum_i Y_i$ and observing that $\mathbb{E}[Y] = p$ as before, we invoke Bernstein's inequality again to obtain that with probability at least $1 - \delta/2$,

$$|Y - p| = \varepsilon_{\mathcal{P}_B} \leq \frac{\log(4/\delta)}{|\mathcal{P}_B|} + \sqrt{\frac{2p(1-p)\log(4/\delta)}{|\mathcal{P}_B|}}.$$

The statement follows by the triangle inequality $|Y - X| \leq |Y - p| + |X - p|$ and the union bound. $\square$

# D    ADDITIONAL EVALUATIONS & EXPERIMENTAL DETAILS

Here, we describe the experimental details and hyperparameters used in our evaluations and provide additional empirical results that supplement the ones presented in the paper. Our additional evaluations support the robustness and widespread applicability of our approach.

## D.1    EXPERIMENTAL DETAILS

We conduct our evaluations on 64 TPU v4s each with two cores. We used a validation data set of size $1,000$ for the CIFAR10, CIFAR100, and SVHN data sets, and used a validation data set of size $10,000$ for ImageNet, respectively, to estimate $m$. The hyper-parameters used with respect to each architecture and corresponding data set(s) are as follows.

**MobileNet (CIFAR10, CIFAR100, SVHN**    For the experiments involving MobileNet ([Howard et al.](), [2017]), whenever MobileNet was used as a student architecture it was initialized with a width paramater of 1, and whenever it was used as a teacher, it was initialized with a width parameter of 2. We used the Adam optimizer ([Kingma & Ba](), [2014]) with default parameters (learning rate: $1\text{e}{-3}$) and trained for either 100 or 200 epochs depending on the experimental configuration. We did not use data augmentation or weight regularization.

**ResNets and ResNetv2s (CIFAR10, CIFAR100, SVHN**    We used the Adam optimizer ([Kingma & Ba](), [2014]) with the default parameters except for the learning rate schedule which was as follows. For a given number of epochs $n_{\text{epochs}} \in \{100, 200\}$, we used $1\text{e}{-3}$ as the learning rate for the first $(2/5)n_{\text{epochs}}$, then used $1\text{e}{-4}$ until $(3/5)n_{\text{epochs}}$, $1\text{e}{-5}$ until $(4/5)n_{\text{epochs}}$, $1\text{e}{-6}$ until $(9/10)n_{\text{epochs}}$, and finally $5\text{e}{-7}$ until then end. We used rounded values for the epoch windows that determine the learning rate schedule to integral values whenever necessary. We did not use data augmentation or weight regularization.

## D.2 IMAGENET

**Setup**    We used a ResNet-18 student and ResNet-50 teacher model for the ImageNet experiments. We train the student model for $100$ epochs using SGD with momentum ($= 0.9$) with batch size $256$ and a learn rate schedule as follows. For the first 5 epochs, we linearly increase the learning rate from $0$ to $0.1$, the next 30 epochs we use a learning rate of $0.1$, the next 30 after that, we use a learning rate of $0.01$, the next 20 we use a learning rate of $0.001$, and use a learning rate of $0.0001$ for the remaining epochs. We use random horizontal flips as our data augmentation.

**Methods**    The implementations of RAD, MARGIN, ENTROPY, and UNIFORM are the same as in our evaluations of CIFAR10/100 and SVHN in Sec 4. However, the Cluster Margin (CM) (Citovsky et al., 2021) and CORESET (Sener & Savarese, 2017) algorithms require expensive clustering operations and were inapplicable to ImageNet off-the-shelf due to memory and computational constraints. Nevertheless, we implemented an approximate version of CM for completeness. We choose CM over CORESET because it is currently the state-of-the-art and Citovsky et al. (2021) have already demonstrated that it outperforms CORESET on large-scale settings. For our approximate version of CM, we partition the $\approx 1.1M$ images into buckets of size $10,000$ and run HAC clustering in each bucket. We stop when the number of generated clusters reaches $500$. This leads to $56,000$ clusters total, after which we apply the CM algorithm in its usual way.

## D.3 BEYOND TEST ACCURACY & COMPARISON TO GREEDY

We investigate the performance of the student model beyond the final reported test accuracy and verify the validity our problem formulation. In particular, we question (i) whether our method also leads to improved student accuracy across all epochs during training and (ii) whether it actually achieves a higher gain with respect to the robust distillation formulation ((4), Sec. 3) compared to using the (greedy) STANDARD MARGIN algorithm that simply picks the highest gain points and UNIFORM sampling. To this end, we conduct evaluations on CIFAR10, CIFAR100, and SVHN data sets similar to those in Sec. 4, and additionally report the test accuracy over each epoch for the last knowledge distillation iteration and the realized gain over the active distillation iterations.

Fig. 5 summarizes the results of our experiments for various knowledge distillation configurations. From the figures, we observe that our approach simultaneously achieves a higher final test accuracy (first column) and generally higher test accuracy over the entire training trajectory (second column). This suggests that the improvements we obtain from our approach are consistent and present regardless of when the student training is terminated. In the third column of Fig. 5, we also observe that our approach achieves the highest realized gain among the evaluated methods and that this gain tends to be a good predictor of the method's performance. This sheds light into why the greedy variant (STANDARD MARGIN) that simply picks the points with the lowest margin (highest gain) is not consistently successful in practice: the high gain points are often mislabeled by the teacher, further confusing the student. This further motivates our robust formulation in Sec. 3 and supports its practicality.

## D.4 APPLYING RAD TO STANDARD ACTIVE LEARNING

Here, we demonstrate the applicability of RAD to standard active learning settings and compare its performance to SOTA strategies. This is motivated by recent work that has demonstrated that selecting the most difficult or informative – with respect to a proxy metric – samples may in fact hinder training of the model (Paul et al., 2022; 2021). For example, on CIFAR10, choosing the most difficult instances was observed to hurt training, and the best strategy was found to be one where moderately difficult points were picked (Paul et al., 2022; 2021). This is method of sampling is reminiscent of the sampling probabilities generated by RAD as depicted in Fig. 2. The results of the experiments are shown in Fig. 6. RAD matches or improves on the performance of state-of-the-art techniques.

The results of the active learning experiments are shown in Fig. 6. Since there is no teacher model involved, we instantiate RAD with $m = 0.05n$ for the number of teacher mistakes. This is based on the empirical studies showing that around $5\%$ of the data points are inherently too difficult (Paul et al., 2022; 2021) or outliers which may impair training. Setting the appropriate value for $m$ when applying RAD to the standard active learning setting remains an open question, and is an interesting

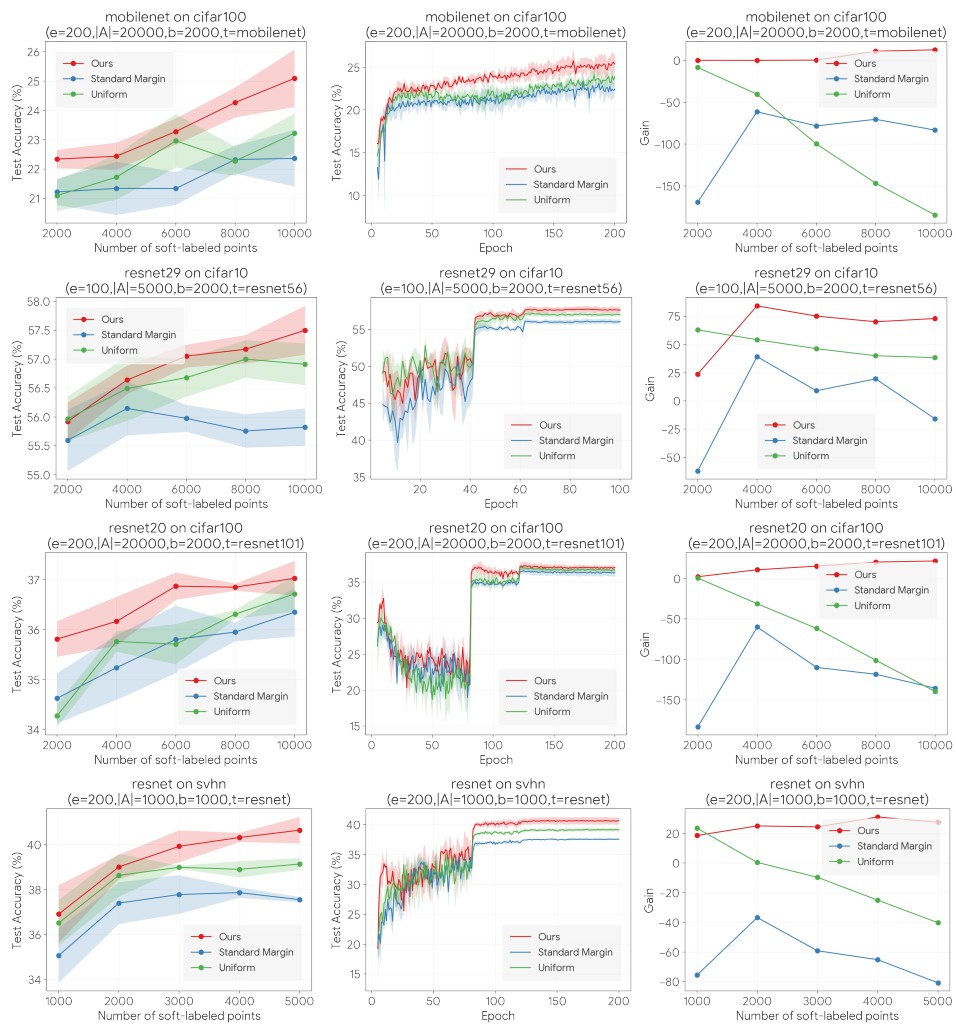

Figure 5: The student's final accuracy (first column), student's test accuracy over the training trajectory (second column), and the realized gains (third column). Our approach generates student models that generally achieve higher test accuracy and gain with respect to the formulation in Sec. 3 across the evaluated scenarios.

direction for future work. The results in Fig. 6 show that RAD is competitive with state-of-the-art active learning algorithms in the evaluated scenarios and matches or improves the performance of the best-performing active learning technique. We emphasize that, in contrast to existing clustering-based approaches such as CM or Coreset, RAD achieves this performance in a computationally-efficient and fully parameter-free way for a given $m$.

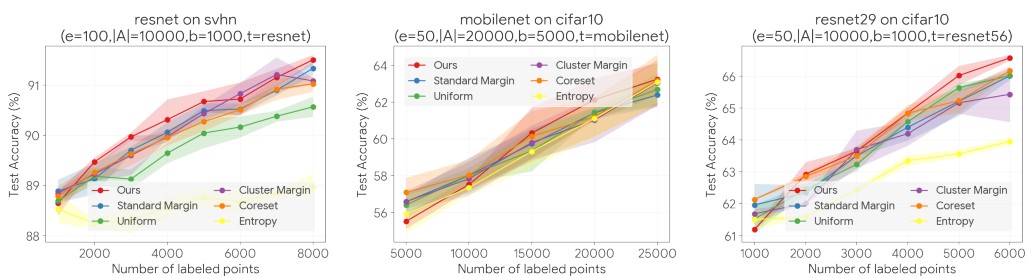

Figure 6: Evaluations in the standard active learning setting.

## D.5 ROBUSTNESS TO THE CHOICE OF $w$

In this subsection, we evaluate the robustness of RAD by evaluating the performance of the algorithm with various instantiations for the $w$ parameter on a wide range of distillation scenarios spanning CIFAR10, CIFAR100, and SVHN datasets and various resnet student-teacher architectures. The results of experiments comparing RAD with the default setting of $w = 1 - m/n$ as described in Sec. 3 (Ours) to RAD variants with $w \in \{0.0, 0.1, 0.2, 0.3, 0.5, 0.6, 0.7, 0.8, 1.0\}$ are shown in Fig. 7. The results were averaged over 5 trials. As we can see from the figure, the performance of RAD remains relatively consistent (generally within one standard deviation) over varying choices of $w$. Moreover, the theoretically-derived choice of $w = 1 - m/n$ consistently performs well across the evaluated scenarios – it is always within one standard deviation of the best-performing $w$ for each scenario.

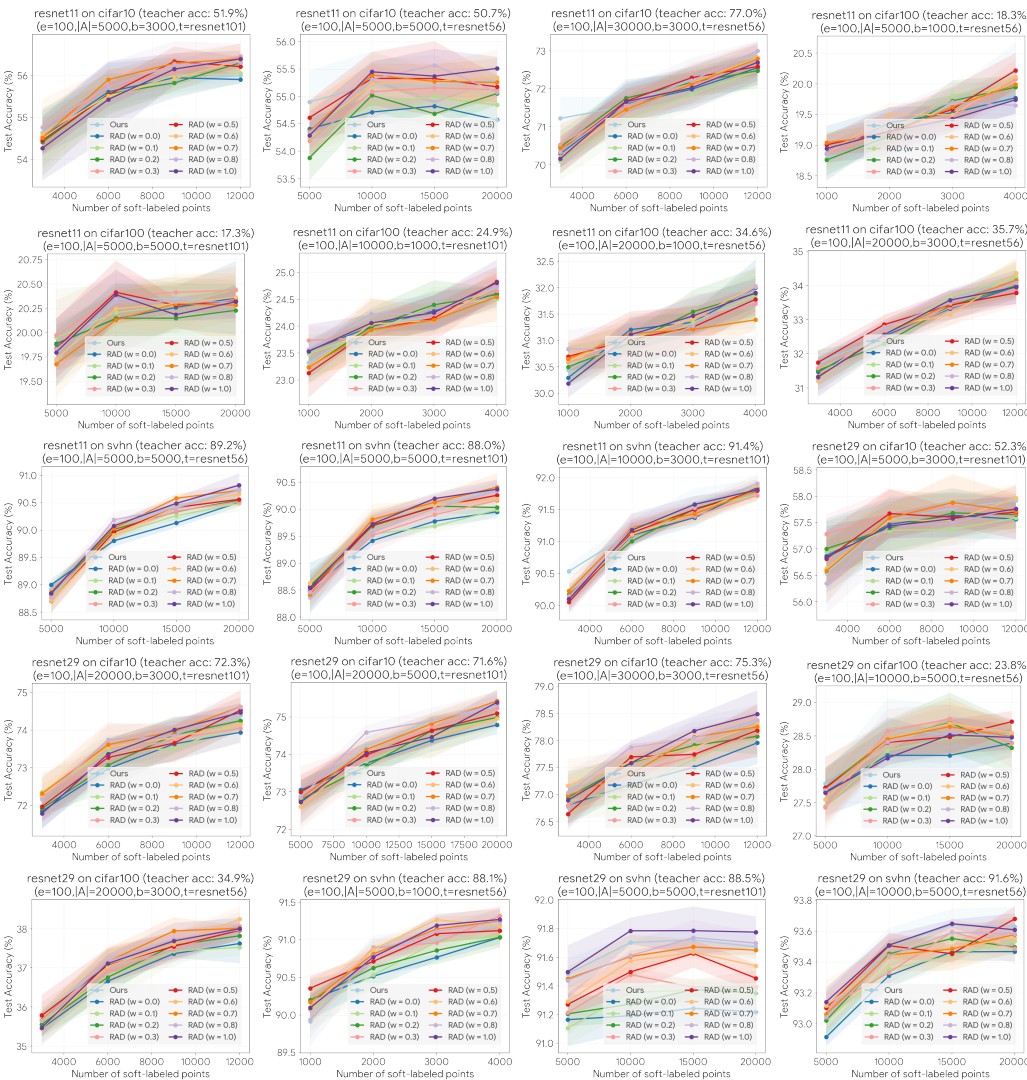

Figure 7: Comparisons of the performance of RAD with various settings of the hyper-parameter $w$ compared to OURS, which uses the default value of $w = 1 - m/n$ (i.e., teacher accuracy). The overlapping performance of the various instantiations (within shaded region of one standard deviation) supports the robustness of RAD to the various settings of $w \in [0, 1]$.

### D.6 ROBUSTNESS TO THE CHOICE OF GAIN

In our empirical evaluations, we had so far only considered a specific definition of gains with respect to the student's margin as described in Sec. 3, i.e., $g_i = 1 - \text{margin}_i$. Since RAD can generally be used with any user-specified notion of gain, in this section we investigate the performance of RAD when the entropy of the student's softmax prediction $f_{\text{student}}(x_i) \in [0, 1]^k$ is used to define the gains, i.e.,

$$g_i = \text{Entropy}(f_{\text{student}}(x_i)) = -\sum_{j=1}^{k} f_{\text{student}}(x_i)_j \log f_{\text{student}}(x_i)_j.$$

We label this algorithm RAD ENTROPY and compare its performance to our variant that uses the student margins.

Fig. 8 shows the results of our comparisons with varying distillation configurations, architectures, and data sets averaged over 5 trials. Overall, we observe that the change in the definition of gain does not lead to a significant change ($>$ one standard deviation) in performance.

### D.7 ROBUSTNESS TO VARYING CONFIGURATIONS

In this section, we consider the robustness of our algorithm to varying configurations on a *fixed data set*. In particular, we consider the SVHN (Netzer et al., 2011) data set and consider the performance with varying size of the student model (ResNetv2-{11, 20, 29}), the size of the teacher (ResNetv2-{56, 110}), $|A| \in \{5000, 10000, 20000\}$, $b \in \{1000, 2000, 5000\}$, and number of epochs $e = \{100, 200\}$. Due to resource constraints, we conduct the extensive comparisons against the top-2 best performing algorithms from the main body of the paper (Sec. 4): (STANDARD) MARGIN and UNIFORM. The results of the evaluations show that our method uniformly performs better or at least as well as well as the competing approaches.

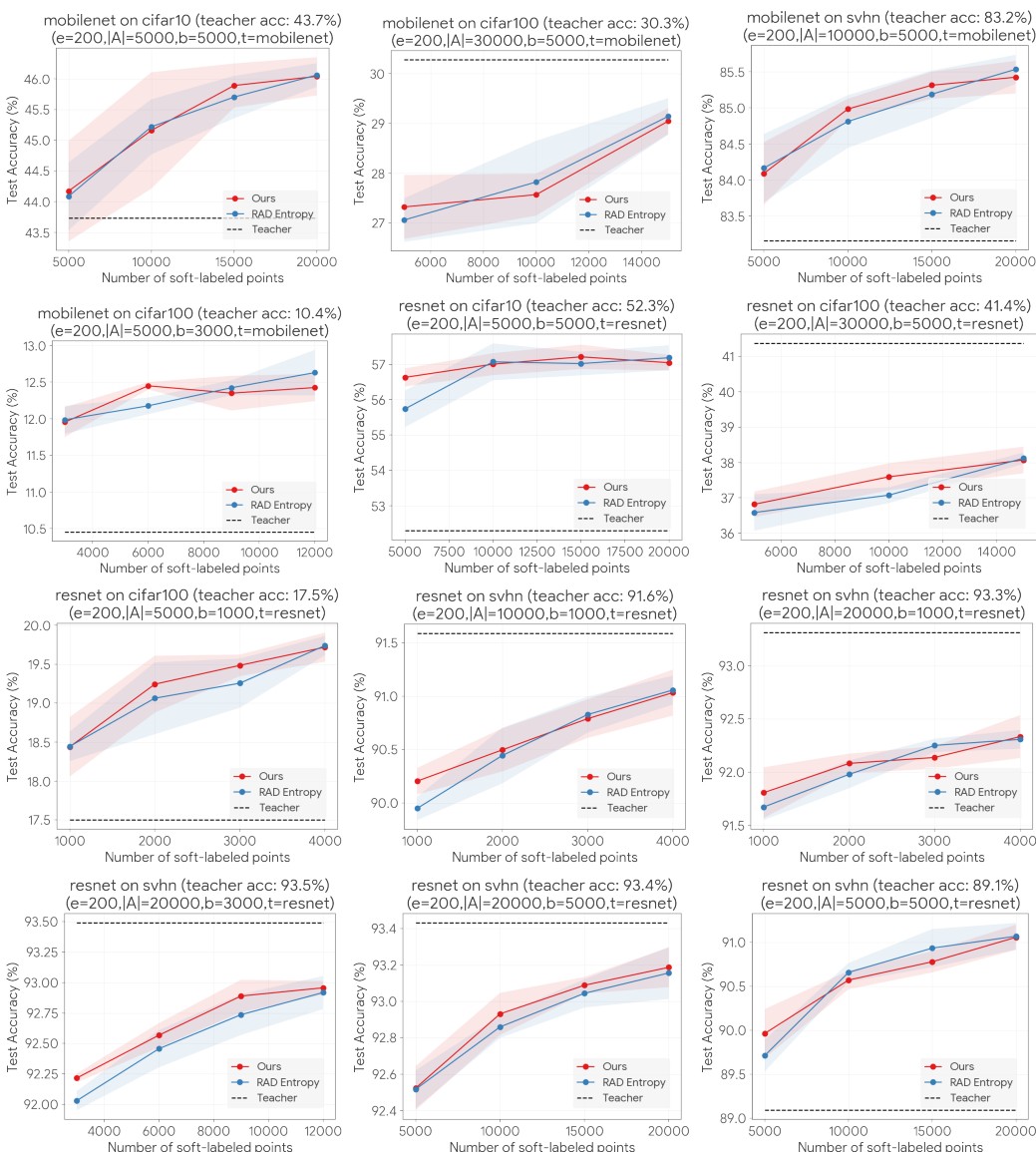

Figure 8: Comparisons of RAD with the gains defined with respect to the student margins as in Sec. 3.1, OURS, to RAD with gains defined with respect to the entropy of the student predictions, RAD ENTROPY. RAD's performance is robust to the alternative notion of uncertainty to define the gains.

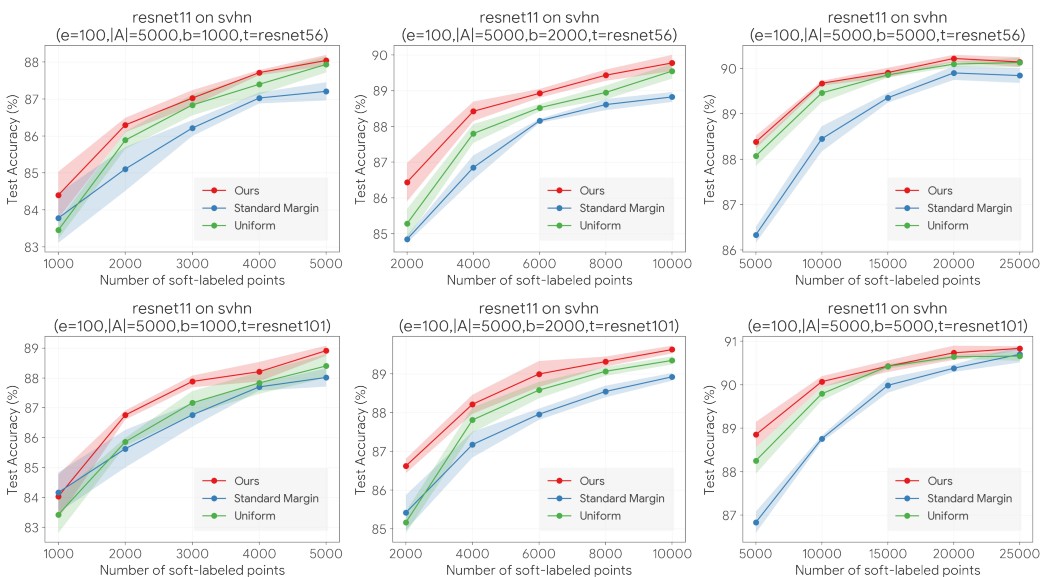

Figure 9: ResNetv2-11 architecture with 100 epochs and $|A| = 5000$. First row: ResNetv2-56 teacher; second row: ResNetv2-101 teacher. Columns correspond to batch size of $b = 1000$, $b = 2000$, and $b = 5000$, respectively.

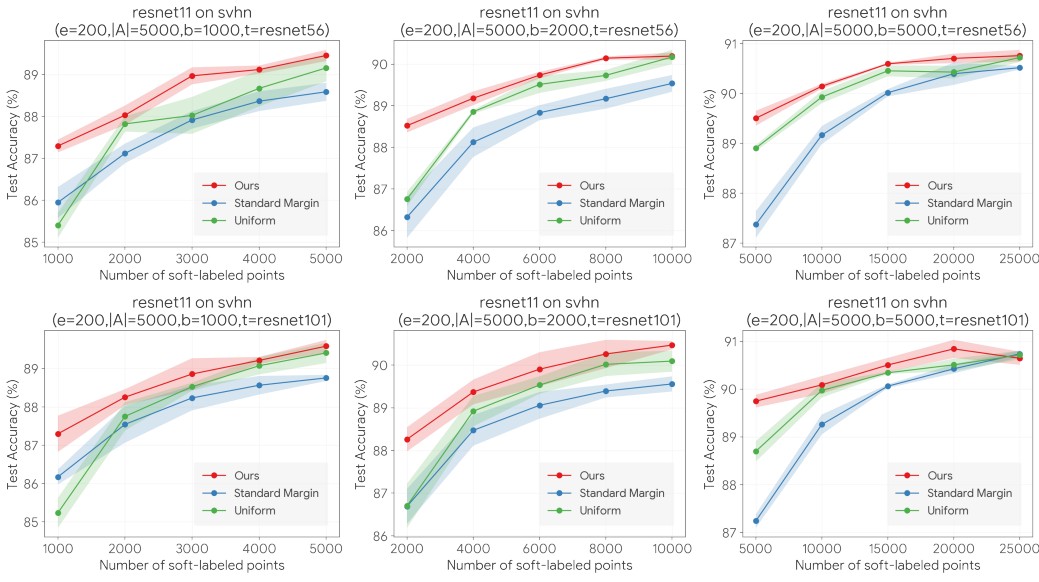

Figure 10: ResNetv2-11 architecture with **200** epochs and $|A| = 5000$. First row: ResNetv2-56 teacher; second row: ResNetv2-101 teacher. Columns correspond to batch size of $b = 1000$, $b = 2000$, and $b = 5000$, respectively.

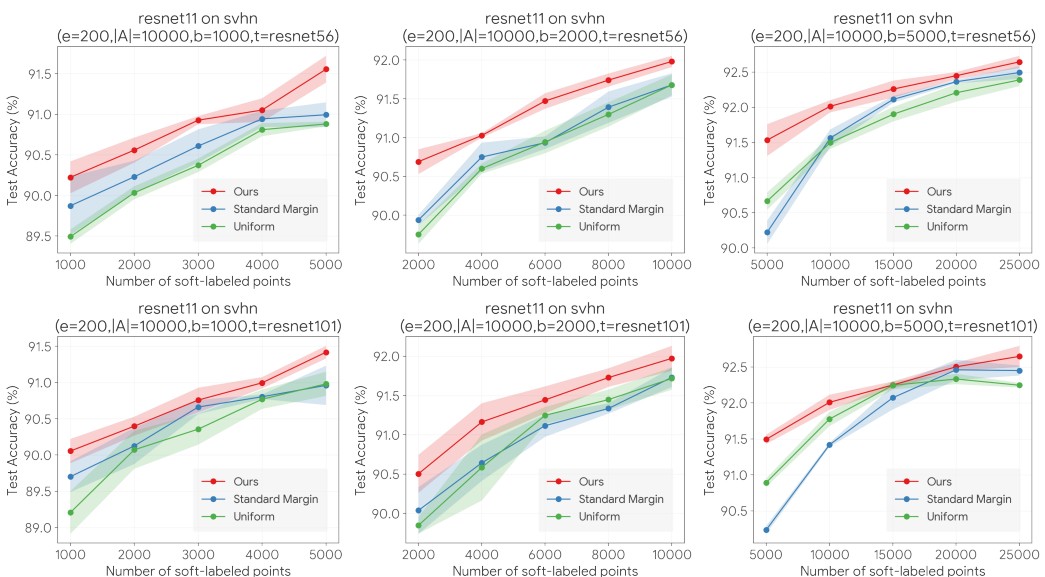

Figure 11: ResNetv2-11 architecture with **200** epochs and $|A| = 10000$. First row: ResNetv2-56 teacher; second row: ResNetv2-101 teacher. Columns correspond to batch size of $b = 1000$, $b = 2000$, and $b = 5000$, respectively.

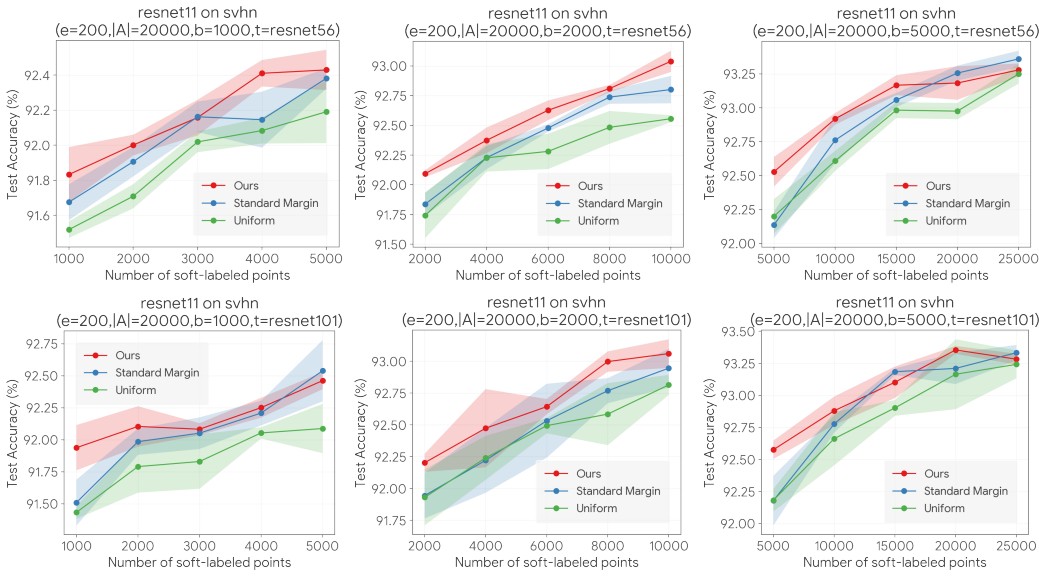

Figure 12: ResNetv2-11 architecture with **200** epochs and $|A| = 20000$. First row: ResNetv2-56 teacher; second row: ResNetv2-101 teacher. Columns correspond to batch size of $b = 1000$, $b = 2000$, and $b = 5000$, respectively.

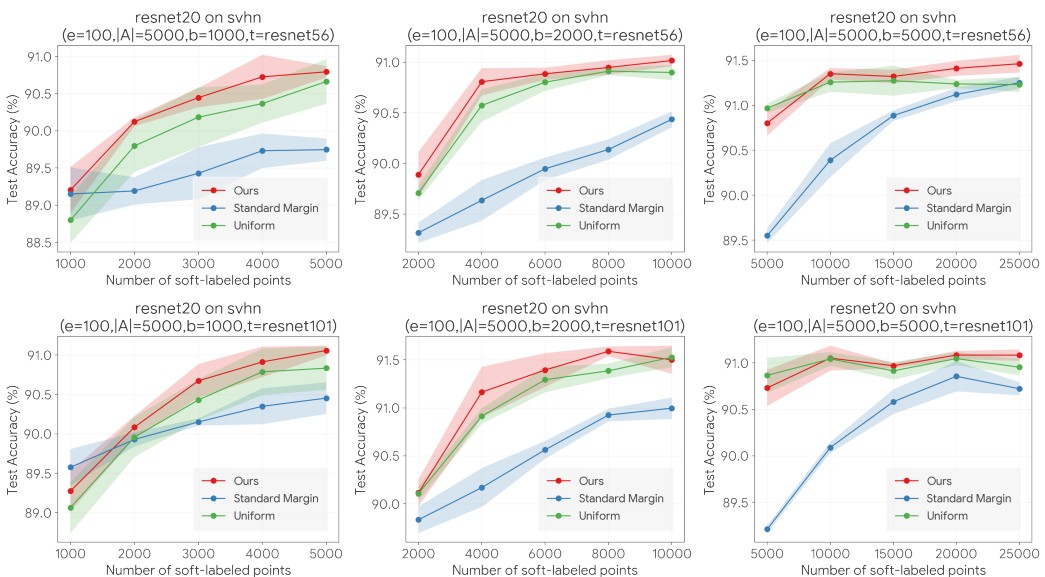

Figure 13: ResNetv2-20 architecture with 100 epochs and $|A| = 5000$. First row: ResNetv2-56 teacher; second row: ResNetv2-101 teacher. Columns correspond to batch size of $b = 1000$, $b = 2000$, and $b = 5000$, respectively.

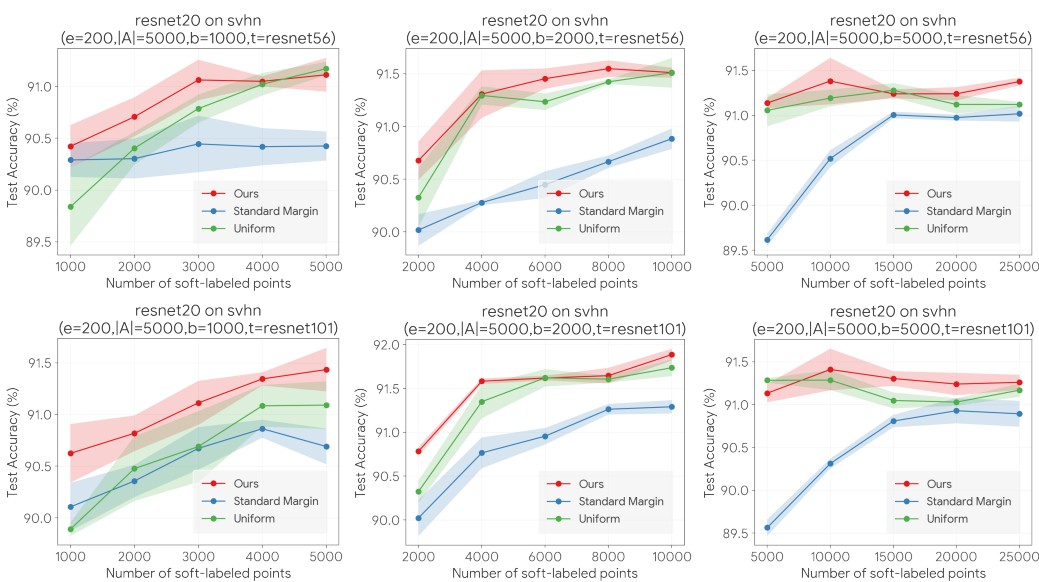

Figure 14: ResNetv2-20 architecture with **200** epochs and $|A| = 5000$. First row: ResNetv2-56 teacher; second row: ResNetv2-101 teacher. Columns correspond to batch size of $b = 1000$, $b = 2000$, and $b = 5000$, respectively.

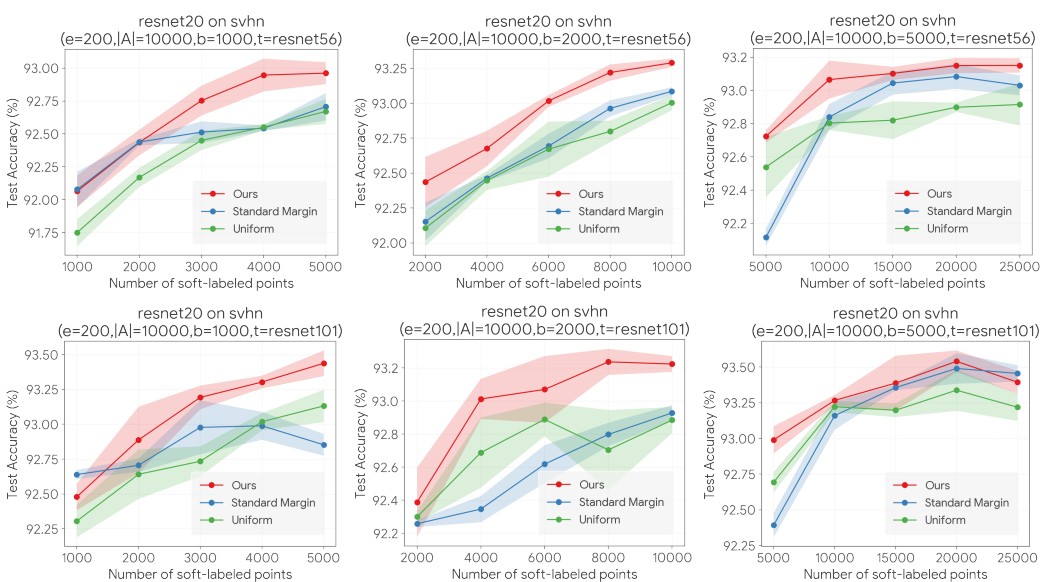

Figure 15: ResNetv2-20 architecture with 200 epochs and $|A| = 10000$. First row: ResNetv2-56 teacher; second row: ResNetv2-101 teacher. Columns correspond to batch size of $b = 1000$, $b = 2000$, and $b = 5000$, respectively.

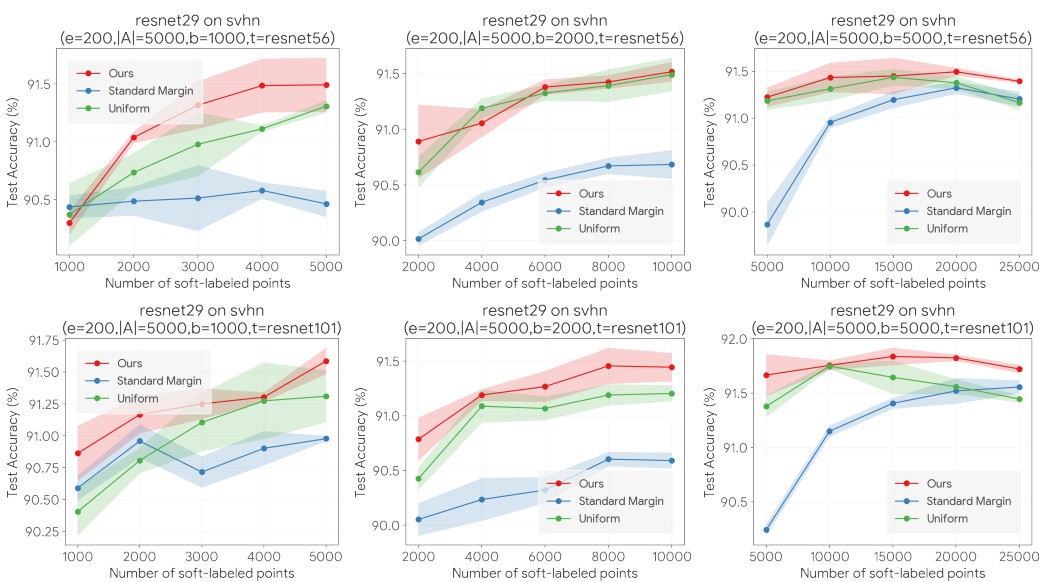

Figure 16: ResNetv2-29 architecture with 200 epochs and $|A| = 5000$. First row: ResNetv2-56 teacher; second row: ResNetv2-101 teacher. Columns correspond to batch size of $b = 1000$, $b = 2000$, and $b = 5000$, respectively.

## D.8 EXPERIMENTS WITH PRE-TRAINED TEACHER MODELS

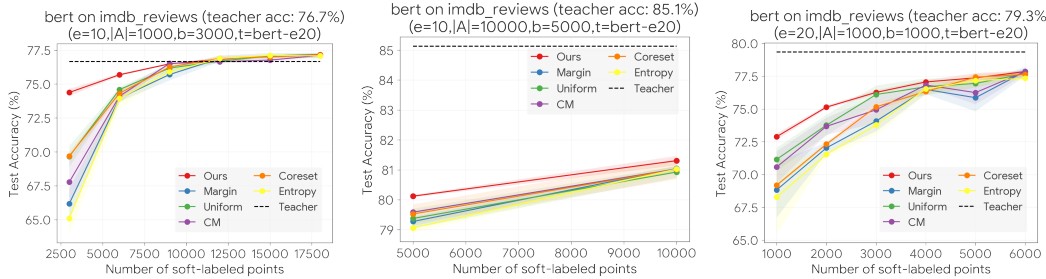

Figure 17: Evaluations on with a ResNet teacher model pre-trained on ImageNet and fine-tuned on the respective data sets. RAD uniformly performs at least as well as the best comparison method, and often better especially in the low sample regime.

In this section, we present evaluations with ResNet50 and ResNet101 teacher models that are pre-trained on ImageNet and fine-tuned on the labeled data that is available for the academic data sets we consider. Fig. 17 depicts the results of our evaluations on CIFAR100 and SVHN datasets. Consistent with the trend of our results in Sec. 4, RAD uniformly outperforms or matches the performance of the best performing comparison method across all scenarios.

## D.9 NLP EVALUATIONS

Figure 18: Evaluations on the IMDB dataset with a tiny BERT student model and a pre-trained BERT teacher.

We conclude the supplementary results by presenting evaluations on a Natural Language Processing (NLP) task on the IMDB Reviews (Maas et al., 2011) dataset with a pretrained BERT teacher model. The IMDB dataset has 25,000 training and 25,000 testing data points, where each data point is a movie review. The task is to classify each review as either positive or negative. We used a pre-trained, 12-layer SmallBERT (Turc et al., 2019) with hidden dimension 768 as the teacher model and a randomly initialized 2-layer SmallBERT with hidden dimension 128 as the student. From Fig. 18 we that the improved effectiveness of RAD relative to the compared approaches persists on the NLP task, consistent with our evaluations on the vision datasets. RAD is particularly effective in the small sample regime, where the number of soft-labeled points is small relative to the size of the unlabeled dataset.

