# OpenReview forum: "Robust Active Distillation"
_ICLR.cc/2023/Conference — ICLR 2023 poster_

### Official Review · Reviewer_fKaY · 2022-10-21

**Confidence:** 4
**Correctness:** 3
**Technical Novelty And Significance:** 2
**Empirical Novelty And Significance:** 3
**Recommendation:** 6

**Clarity, Quality, Novelty And Reproducibility:**

This paper is generally well-written and easy to follow. The minimax formulation to simultaneously achieve efficiency and robustness seems to be quite novel to me. The authors also provide details (e.g., hyperparameters) to reproduce their results.

**Strength And Weaknesses:**

Strength:
1. The authors proposed an interesting minimax optimization problem to select data points, which ensures both efficiency and robustness. The idea behind this optimization problem seems novel to me.
2. The authors identify a theoretically-motivated weight parameter such that the expected gain is never negative. As a result, the proposed algorithm can be operated in a parameter-free manner.


Weaknesses:
1. The minimax problem seems to be only solved in the case when b=1 (in Thm 1). When the budget is larger than 1, it seems that the authors simply multiply the solution (obtained when b=1) by the budget. Can we directly solve the minimax optimization problem when the budget is greater than 1?
2. The choice of m is estimated based on sample inaccuracy, which seems to require hard-labeled data points. Are there ways to get around with this?

Questions:
1. In algorithm 1, why both the teacher and the student are trained on the same hard-labeled data points? I thought the teacher should be trained on a much larger dataset in order to provide soft-labels that are better than the ones generated by the student.
2. Following the above question, can you provide more explanations on why the student can eventually outperform the teacher in your experiments?

**Summary Of The Paper:**

This paper studies robust active distillation that can simultaneously achieve efficiency and robustness. The query rule of the algorithm follows from the solution of a minimax problem that tries to maximize the "gain" given a limited number of mislabels. The minimax optimization problem can be solved in nearly linear time (with closed-form solutions provided in Thm 1 and Cor 1).  The authors conduct extensive experiments which empirically confirm the efficacy of their proposed method.

**Summary Of The Review:**

Based on the detailed reviews above, I think this paper has proposed some interesting and novel ideas. I also would like to hear the author's rebuttal regarding the weaknesses/questions.

====after rebuttal====

Thank you for your response. I would like to keep my current scores. I believe the paper can be made stronger if the authors can provide some analyses for the case with b>1.

---

> ### Author Response · Authors · 2022-11-17
> **(Part 1/2): Response to Reviewer fKaY**
>
> We are grateful for the reviewer’s careful consideration of our paper and for their insightful comments. Our specific responses are below.
>
> ### Questions in the ‘Weaknesses’ section
> 1. You are correct that we solve the problem for $b = 1$ and then obtain a solution for the batch sampling case by multiplying by $b$. As we state in our submission, this is optimal as long as the probabilities are not concentrated on a single point and $b$ is not too large. This is a very mild assumption given the inherent spread of the probability distribution that RAD provides (Fig. 2) and the setting that we consider, where the number of unlabeled points is much larger than $b$, i.e., $|X_U| \gg b$.
>
>     We agree that the case of $b>1$ is mathematically interesting. In our work, we were able to generate an optimal solution for $b>1$ for certain cases. The algorithm was significantly more complicated, and empirically, its performance was virtually indistinguishable from the approach we presented in the paper – since the solutions only differed when $\max_{i} p_i > 1/b$, which did not occur in our evaluations. However, this is an interesting line for future research, and we are considering the optimal solution for the $b>1$ case in full generality for future work.
>
> 2. Thank you for your insightful suggestion. Given the current formulation that explicitly takes into account the teacher’s accuracy, we believe that leveraging some information about the teacher's accuracy on a problem-dependent basis is a feature and not a constraining assumption. After all, the points that we select to be soft-labeled should in some way depend on whether we are dealing with a high quality (accurate) teacher whose soft-labels we can trust or not. Moreover, a validation data set (containing points with hard labels) is available in virtually all ML applications to, e.g., tune hyperparameters and/or for early stopping, and this very same set can be used to approximate the value of $m$. Even if $m$ is approximate due to a small validation data set (we used a validation set of size 1,000 for our experiments),  our theoretical and empirical results in the appendix indicate that our method still works well. We clarified this in our revision.

---

> > ### Author Response · Authors · 2022-11-17
> > **(Part 2/2): Response to Reviewer fKaY**
> >
> >
> > ### Remaining questions
> >
> > 1. The teacher and student were trained on the same labeled data set under the semi-supervised KD observation that a higher capacity model is more capable of making better use of limited data. Therefore, even if the teacher is not pre-trained, the premise is that it can effectively provide informative soft-labels for the unlabeled points due to the sheer capacity of the teacher model relative to that of the student.
> >
> >     Nevertheless, in many settings, the teacher is pre-trained and a small, domain-specific labeled data set is used to fine-tune the teacher. To show that RAD performs well even in the pre-trained setting, we conducted additional evaluations in settings involving pre-trained teacher models (as we mention in our General Response). In particular, Fig. 17 of our revision presents evaluations using pre-trained (on ImageNet) ResNet models on CIFAR100 and SVHN datasets. Additionally, Fig. 18 shows the results of experiments on an NLP task using a pre-trained teacher model. We see that RAD is uniformly effective in this setting, similar to the results in the setting of our original submission.
> >
> > 2. We agree that it is seemingly counterintuitive that the student model’s test accuracy can exceed that of the teacher model. However, this mysterious phenomenon has been observed and validated by other works in knowledge distillation, even in the fully-supervised setting [1-5]. In fact, prior work has found that self-distillation, where the teacher and student have the same network architecture, can lead to student models that outperform the teacher model (see, e.g., [1,5,6]). This effect is more pronounced in the semi-supervised KD setting that is the focus of our work, since the student model is trained on more data than the teacher model: the labeled examples and the teacher’s soft labels on a subset of the unlabeled set. The most relevant work in this realm is [3], which considers the semi-supervised KD setting. They too report results where the student model outperforms the teacher, especially as the student model is trained on increasingly more data soft-labeled by the teacher (e.g., see Figs. 4-8 and Figs. 9-10 of [3]).
> >
> >     To supplement the discussion above, the new evaluations in our revision (Figs. 17 and 18) with a high capacity pre-trained teacher model provide additional examples where the student’s performance is virtually always below that of the teacher.
> >
> >
> > We would be happy to answer any additional questions. Thank you again for your constructive feedback and consideration.
> >
> >
> > [1]  https://arxiv.org/pdf/1911.04252.pdf
> >
> > [2] https://arxiv.org/pdf/2210.06711.pdf
> >
> > [3] https://arxiv.org/pdf/2005.10419.pdf
> >
> > [4] https://arxiv.org/pdf/2012.09816.pdf
> >
> > [5] https://arxiv.org/pdf/2106.05945.pdf
> >
> > [6] https://arxiv.org/pdf/1905.08094.pdf

---

### Official Review · Reviewer_26jz · 2022-10-22

**Confidence:** 3
**Correctness:** 4
**Technical Novelty And Significance:** 4
**Empirical Novelty And Significance:** 3
**Recommendation:** 8

**Clarity, Quality, Novelty And Reproducibility:**

The paper is well written overall and presents an interesting idea supported by solid experimental results.

The paper can use a proofread to fix minor presentation issues such as: What is X_B in line 2 of Section 2.1?

**Strength And Weaknesses:**

Strengths:

1. The proposal that considers the problem of importance sampling for simultaneous efficiency and robustness in knowledge distillation is novel and interesting.

2. The proposal is supported by non-trivial theoretical analysis.

3. The proposed techniques are shown to be effective on multiple real datasets.

Weakness:
The paper motivates with KD using large NLP models such GPT-3. It would be good to incorporate experiments on NLP datasets and tasks.



**Summary Of The Paper:**

This paper proposes an active learning-based approach for knowledge distillation. It assumes that the teacher model does not have access to a fully labelled dataset either, and there is a cost associated to every call of the teacher model to produce soft labels for the unlabelled data. The aim is to minimize the total cost of making teacher model inference calls while maximizing the student accuracy learned from the soft labels produced by the teacher model. Its core idea is to select the samples where the teacher model is more confident instead of those where the student model is most unconfident. Experimental results on multiple real datasets confirm the effectiveness of the proposed technique.

**Summary Of The Review:**

The paper considers the problem of importance sampling for simultaneous efficiency and robustness in knowledge distillation which is an interesting idea. The proposed solution is supported by non-trivial theoretical analysis and strong results on real datasets. Overall, the paper presents a solid contribution that has the potential to intrigue follow-up studies.

=== Update after rebuttal ==

Thanks for the response. I have no further questions.

---

> ### Author Response · Authors · 2022-11-17
> **Response to Reviewer 26jz**
>
> We would like to thank the reviewer for their thoughtful consideration and positive feedback on our work. We have revised the manuscript and fixed all typos.
>
> Thank you for your suggestion regarding an NLP evaluation. As we mention in our General Response, we included evaluations on an NLP task with a pre-trained BERT model in our revision (Fig. 18 in the appendix). Consistent with our results on the vision datasets, RAD outperforms or matches the performance of the best baseline method. We plan to conduct more extensive NLP evaluations in future work. Thank you for helping us improve our work.

---

### Official Review · Reviewer_FQ16 · 2022-10-24

**Confidence:** 3
**Clarity, Quality, Novelty And Reproducibility:** The paper is well-written and the pro…
**Correctness:** 4
**Technical Novelty And Significance:** 3
**Empirical Novelty And Significance:** 3
**Recommendation:** 6

**Strength And Weaknesses:**

Strength:

1. The author presented a mathematical formulation that captures the objective of training on informative soft labels that are accurately labeled by the teacher.

2. The solution takes a simple and intuitive game-theoretical viewpoint, which can be obtained in near-linear time without tuning the hyperparameter.

3. The author also presents extensive empirical evaluations that demonstrate the effectiveness of the proposed framework.

Questions and Weaknesses:

1. At the beginning of page 3, the author state that "a very limited number of ground truth label queries can be made for the points in $X_i$ throughout the distillation process". From my understanding, the ground truth label queries are different from querying the teacher model for soft labels. How exactly are those ground truth label queries used?

2. The game-theoretic approach can deal with very bad quality soft-label, which seems somewhat pessimistic. Is there any way we can estimate the quality of the soft-label without using oracle? For instance, what will happen if we use the margin of the teacher-generated soft-label as a proxy for $c_i$ in equation (2) ?


**Summary Of The Paper:**

In this paper, the authors presented a model distillation method that can actively select the training samples for distillation. The method is efficient and robust - in the sense that the method suffers less from inaccurate soft labels. The proposed method is parameter-free. The authors further verified the efficacy of the proposed method empirically.

**Summary Of The Review:**

This paper proposed an intuitive, interesting and effective approach for active distillation. With the theoretical understanding and the extensive empirical results, I think this paper is above the acceptance threshold.

---

> ### Author Response · Authors · 2022-11-17
> **Response to Reviewer FQ16**
>
> Thank you for your supportive review and helpful suggestions. Please see our responses to questions 1 and 2 below.
>
>
> 1. The ground truth label queries are only used as a validation data set to measure the teacher’s test accuracy, i.e., they are only used to approximate the number of mistakes $m$ that is given as input to RAD. In retrospect, we fully agree that this wording may be a source of confusion and we have revised this in our revision to more simply state that we assume a validation data set is available, as is standard in ML applications.
>
> 2. Thank you for your insightful suggestion. We agree that trying to approximate the $c_i$’s is an appealing idea. The only catch here is that we cannot use the teacher’s margin scores to approximate or predict the $c_i$’s because computing the margin scores of the unlabeled points requires the teacher’s prediction (soft-label) for every unlabeled point. This is not feasible due to the soft-labeling budget imposed by the computational and/or financial costs of running inference with the teacher model.
>
>     With a black-box teacher model that only exposes the teacher’s soft-labels and not internals of the model itself, we can only hope to leverage the student model to predict the $c_i$’s. This is an approach that we tried prior to developing RAD, but found it very difficult to reliably predict whether the teacher would label a point correctly or not. In fact, the unreliability of these predictions was one of the motivating factors for RAD. Overall, we fully agree that leveraging predictions in a clever way and accounting for the possible prediction inaccuracies may result in a less pessimistic approach. We think this is a promising avenue for future work and thank the reviewer again for their helpful suggestion.
>
> We would be happy to answer any further questions that you may have.

---

### Author Response · Authors · 2022-11-17
**General Response**

We would like to thank the reviewers for their careful consideration and constructive feedback of our work.

In light of the reviewers’ comments, we uploaded a revision that, to the best of our knowledge, addresses all of the reviewers’ feedback. We highlighted the major additions in blue for visibility. In addition to an improved exposition, the revision contains


1. Supplementary evaluations on CIFAR100 and SVHN with pre-trained (on ImageNet) teacher models, as recommended by reviewer fKaY. Please see the last page of the supplementary for the results (Fig. 17).
2. Evaluations on an NLP task with a pre-trained BERT model as the teacher, as suggested by reviewer 26jz. Please see the last page of the supplementary for the results (Fig. 18).

The trend in these additional evaluations is consistent with that of our original results: RAD uniformly outperforms or matches the performance of the best compared method.

We would like to thank the reviewers again for their time and helpful suggestions that have helped improve our work. We look forward to answering any remaining questions during the discussion period.

---

### Decision · Program_Chairs · 2023-01-20

**Decision:**

Accept: poster

**Justification For Why Not Higher Score:**

The submission is not more competitive since its problem setting (i.e., large-scale knowledge distillation) is very specific.

**Justification For Why Not Lower Score:**

All reviewers agreed that the novelty and significance are above the bar.

**Metareview: Summary, Strengths And Weaknesses:**

The paper studied large-scale knowledge distillation about the trade-off between its efficiency (i.e., the number of soft-labels to be queried to the teacher model) and robustness (the accuracy of soft-labels provided by the teacher model). A theoretically guaranteed method was proposed to query the soft-labels of data that are simultaneously informative and correctly labeled by the teacher model. The authors did a good job in the rebuttal to successfully address the concerns from the reviewers. All reviewers agreed that the novelty and significance are above the bar and thus we should accept it for publication.

**Note From Pc:**

if the above contains the word "oral" or "spotlight" please see: "oral" presentation means -> notable-top-5% and "spotlight" means -> notable-top-25%. As stated in our emails, we are disassociating presentation type from AC recommendations